# Social bonding in groups of humans selectively increases inter-status information exchange and prefrontal neural synchronization

Jun Ni[1,2,3], Jiaxin Yang[1,2,3], Yina Ma[1,2,3,4] *

**1** State Key Laboratory of Cognitive Neuroscience and Learning Beijing Normal University, Beijing, China, **2** IDG/McGovern Institute for Brain Research, Beijing Normal University, Beijing, China, **3** Beijing Key Laboratory of Brain Imaging and Connectomics, Beijing Normal University, Beijing, China, **4** Chinese Institute for Brain Research, Beijing, China

* yma@bnu.edu.cn

**Data Availability Statement:** All data needed to reproduce the conclusions and figures in the paper are present in S1 Data and the public repository of Open Science Framework (OSF, https://osf.io/jdzca/). The toolbox and codes of main data

## Abstract

Social groups in various social species are organized with hierarchical structures that shape group dynamics and the nature of within-group interactions. In-group social bonding, exemplified by grooming behaviors among animals and collective rituals and team-building activities in human societies, is recognized as a practical adaptive strategy to foster group harmony and stabilize hierarchical structures in both human and nonhuman animal groups. However, the neurocognitive mechanisms underlying the effects of social bonding on hierarchical groups remain largely unexplored. Here, we conducted simultaneous neural recordings on human participants engaged in-group communications within small hierarchical groups ($n = 528$, organized into 176 three-person groups) to investigate how social bonding influenced hierarchical interactions and neural synchronizations. We differentiated interpersonal interactions between individuals of different (inter-status) or same (intra-status) social status and observed distinct effects of social bonding on inter-status and intra-status interactions. Specifically, social bonding selectively increased frequent and rapid information exchange and prefrontal neural synchronization for inter-status dyads but not intra-status dyads. Furthermore, social bonding facilitated unidirectional neural alignment from group leader to followers, enabling group leaders to predictively align their prefrontal activity with that of followers. These findings provide insights into how social bonding influences hierarchical dynamics and neural synchronization while highlighting the role of social status in shaping the strength and nature of social bonding experiences in human groups.

## Introduction

Most social groups, from the basic family unit to professional organizations, and societal institutions, are hierarchically structured [1,2]. The hierarchical structure and different status relationships are one of the most fundamental features of social groups and shape interpersonal

analysis are available in OSF website https://osf.io/jdzca/.

**Funding:** This work was supported by the National Natural Science Foundation of China (https://www.nsfc.gov.cn/english/site_1/index.html, Projects 32125019 to Y.M.); the STI 2030—Major Projects 2022ZD0211000 to Y.M. (https://en.most.gov.cn/); the Fundamental Research Funds for the Central Universities (http://en.moe.gov.cn/, 2233300002 to Y.M.); the Major Project of National Social Science Foundation (http://www.nopss.gov.cn/, 19ZDA363 to Y.M.); the start-up funding from the State Key Laboratory of Cognitive Neuroscience and Learning, IDG/McGovern Institute for Brain Research, Beijing Normal University (https://brain.bnu.edu.cn/English/index.htm, to Y.M.). The funders had no role in study design, data collection and analysis, decision to publish or preparation of the manuscript.

**Competing interests:** The authors have declared that no competing interests exist.

**Abbreviations:** BOLD, blood oxygenation level dependent; FDR, false discovery rate; fNIRS, functional near-infrared spectroscopy; FOI, frequency band of interest; FWE, family-wise error; IDG, intergroup dictator game; INS, inter-brain neural synchronization; IPD-MDG, intergroup prisoner's dilemma-maximizing differences game; LMM, linear mixed model; MU, monetary unit; rDLPFC, right dorsolateral prefrontal cortex; rTPJ, right temporal-parental junction; WTC, wavelet transform coherence.

interactions among group members [3,4] to facilitate group stability [5] and maximize group productivity [6]. However, hierarchical structure comes with challenges and costs for social groups [7]. In hierarchical groups, high-ranking individuals may bully subordinates and usurp a disproportionate share of resources, social influence, and reproductive opportunities [8,9], which may amplify intragroup inequality and competitions [10], undermine the authority and legitimacy of group leaders [7]. Small groups overcome these problems through in-group social bonding [11,12], an adaptive means of forming, strengthening, and maintaining interpersonal connections with in-group members [13–15]. Social bonding exercises, such as grooming behaviors in nonhuman primate, collective rituals, traditions, and team-building activities in human society [16–19], have been widely adopted to facilitate leader influence [20], increase group cohesion [21], and reinforce the hierarchical structure [22]. Yet, despite the significance and impact of in-group social bonding, the neurocognitive mechanisms underlying the effects of social bonding on hierarchical groups remain largely unknown.

Specifically, we asked how social bonding facilitated interpersonal interactions within hierarchical groups and examined here at both the behavioral and neural levels. The hierarchical structure places individuals at different positions in the group (i.e., individuals with different social statuses), such as group leader and followers, varying in levels of resources, social influence, or competence [10,23]. Sensitivity to status information and recognizing one's relative social status in the group are essential for successful social interaction, and interpersonal interaction within a hierarchical group is shaped by different status relationships [3]. Interpersonal interactions within a simply structured hierarchical group are thus classified into 2 fundamental, status-related types: (i) interactions between individuals of different social status, e.g., leader–follower interaction (henceforth inter-status interaction); and (ii) interactions between individuals of the same social status, e.g., follower–follower interaction, the most common representative of intra-status interactions. We further asked whether and if so, how social bonding differentially influenced these 2 types of interpersonal interactions within a hierarchical group. Previous studies have demonstrated distinct functions of inter-status and intra-status interactions within hierarchical groups [24,25]. Inter-status interaction facilitates the exchange of asymmetric information between the group leader and followers [26], as well as leader–follower coordination and alignment [27]. On the other hand, intra-status interaction fosters reciprocal relationships and peer support among individuals with similar social status (e.g., followers) [1,28]. Therefore, we expected that social bonding would exert different effects on inter-status and intra-status interpersonal interactions within a hierarchical group. Specifically, we examined whether social bonding would exhibit stronger or weaker effects or even opposite effects on inter- versus intra-status interactions.

At the behavioral level, we tested whether social bonding facilitated inter-status interaction, intra-status interaction, or both, and if so, whether such bonding effect on intra- and/or inter-status interaction was linked to the facilitation of in-group cohesion and leader's influence. Moreover, in-group social bonding may increase not only in-group cohesion, but also hostility towards out-group members [15,29,30], resulting in increased intergroup discrimination. We thus also examined how in-group social bonding influenced intergroup discrimination, especially for individuals with different social status (i.e., group leader and followers). At the neural level, we employed functional near-infrared spectroscopy (fNIRS) hyper-scanning to measure neural synchronization among group members. Recent neuroscience research has suggested inter-brain neural synchronization (INS) as a reliable indicator of social interactions [31–33]. Neural synchronization emerges in a variety of social encounters, including interactions between peers [34], romantic partners [35], parent and child [36,37], teacher and student [38]. The degree of neural synchronization was predictive of interaction quality [39]. Of particular interest, neural synchronization among group members has been suggested as a candidate

mechanism mediating within-group interaction [15,40], and in-group bonding increased within-group neural synchronization [15,41]. However, previous studies merely focused on the egalitarian group, suffering from not being able to differentiate inter- and intra-status interactions and leaving the social bonding effect on hierarchical interactions an open question. Here, we aimed to reveal whether and how social bonding influenced neural synchronization within a hierarchical group and, in particular, the inter-status and intra-status neural synchronization. Recent research has documented stronger neural synchronization during social interactions between individuals with different social roles than those with the same roles (e.g., teacher–student versus student–student, [42,43]; leader–follower versus follower–follower, [44,45]). Therefore, it could be expected that in-group social bonding would have differential effects on inter-status and intra-status neural synchronizations.

To address these questions, we applied fNIRS hyper-scanning to 176 three-person groups (the most basic hierarchical group with 1 leader and 2 followers, S1 Table) and simultaneously recorded neural activities of 3 group members of each group during online within-group interactions. Participants democratically elected a group leader and discussed group strategies for potential intergroup contests after in-group social bonding or no-bonding control manipulation (Figs 1A and S1). The in-group social bonding manipulation employed in the current study was adapted from several previously validated procedures [15,46–49]. Specifically, we integrated 3 fundamental procedures to manipulate in-group social bonding: (i) shared preference [46]; (ii) symbolic marker [47]; and (iii) similarity among group members [48]. Taking advantage of fNIRS technology that provides noninvasive measures of neural activity with minimal sensitivity to motion artifacts [50], we measured neural synchronization in the right dorsolateral prefrontal cortex (rDLPFC) and right temporal-parental junction (rTPJ) in inter-status and intra-status dyads (Fig 1B). Brain regions of interest in the current study included rDLPFC and rTPJ. Previous studies have shown that the right (but not left) DLPFC was involved in allocating attention and making strategic decision during social interaction [51,52]. The TPJ, particularly in the right hemisphere, is a core region of the mentalizing network and involved in alignment with in-group members regarding consensus decision and group norms [53]. This ROI choice was also based on earlier studies that have linked neural synchronization in the rDLPFC and rTPJ with social interactive processes [32,36,45,54]. Moreover, neural synchronization in the rDLPFC activity predicted in-group cooperation during intergroup conflict [15], while neural synchronization in rTPJ was associated with leader–follower interaction [45].

## Results

### In-group social bonding facilitated inter-status communication and cohesion

We performed conversation analysis [55] on the transcripts of within-group communication for each group. The within-group communications were operationalized on a turn-taking basis (Fig 1C). Therefore, we focused on the number of utterances, the number of turn transition, and turn response time. Turn transition refers to the exchange of utterances among different group members. The number of turn transitions is suggested to reflect the frequency of mutual understanding and engagement [56], with more turn transitions indicating more interactive and engaging communications between group members. Turn response time is measured by the time interval between turns, with faster response times reflecting stronger social connection and more efficient and interactive communication [57].

We first compared the number of utterances of group leader and followers between social bonding and no-bonding control conditions (Methods). There was a significant main effect of

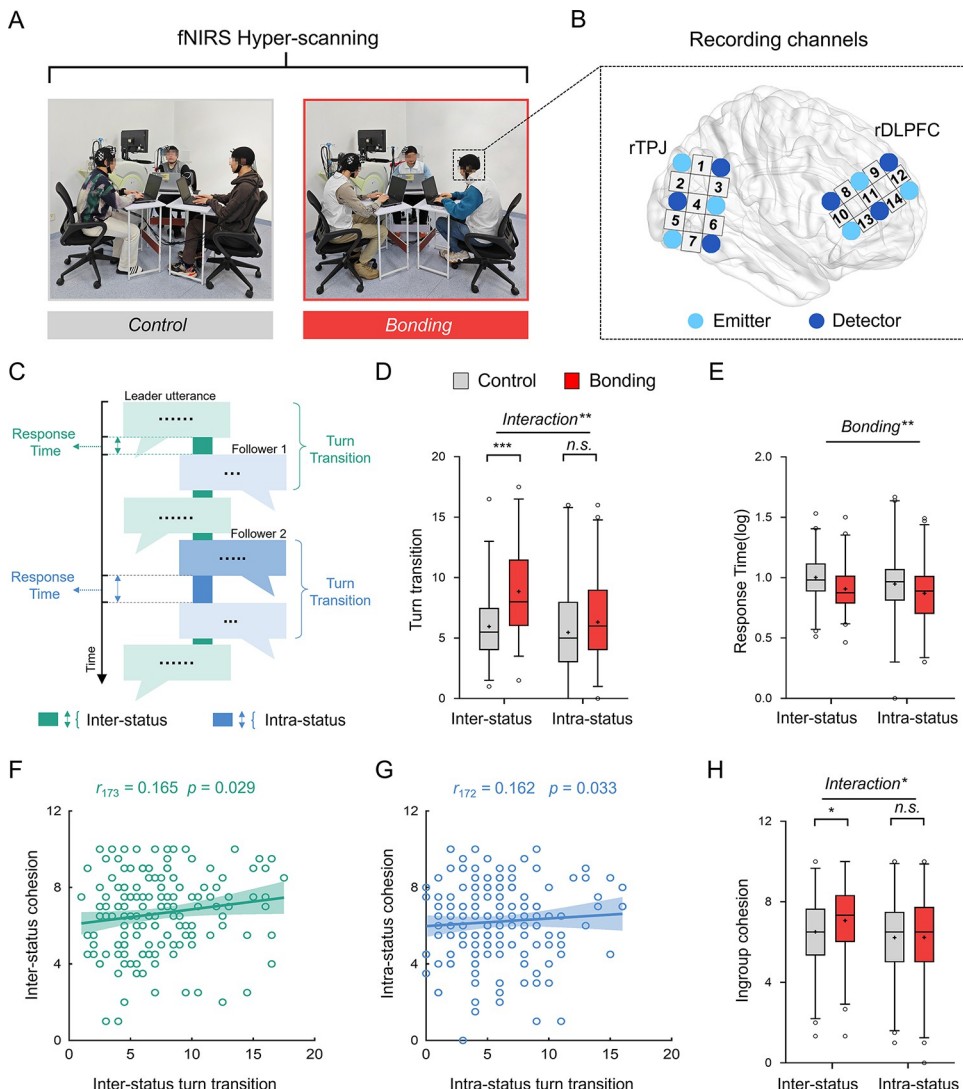

**Fig 1. In-group social bonding selectively facilitates inter-status communication and cohesion.** (**A**) Experimental setting. During the group interaction, 3 group members' rDLPFC and rTPJ signals were simultaneously recorded by the same fNIRS system. Shown are snapshots of a control session and a bonding session. (**B**) Illustration of fNIRS probe configurations. Two identical 3 × 2 optode probe sets, with each consisting of 3 emitters (light blue) and 3 detectors (dark blue), were placed on rDLPFC and rTPJ, respectively. The integers in between indicate the recording channels. (**C**) Group members were engaged in turn-taking conversation session. Each turn transition was defined as a discrete pair of utterances from different individuals (depicted by rectangles). The turn-response time was calculated as the time interval of corresponding turn transition. (**D**) Social bonding increased inter-status turn transitions (control: 5.960 ± 2.971, bonding: 8.854 ± 3.759) but not intra-status ones (control: 5.470 ± 3.560, bonding: 6.320 ± 3.499). (**E**) Bonding significantly sped up turn transitions (inter-status: control: 0.997 ± 0.188, bonding: 0.898 ± 0.172; intra-status: control: 0.947 ± 0.284, bonding: 0.909 ± 0.228). (**F/G**) Interpersonal cohesion was positively associated with turn transition frequency, respectively, for inter-status (**F**) and intra-status dyads (**G**). Each solid line represents the least squares fit, with shading showing the 95% CI. (**H**) Bonding selectively increased inter-status (control: 6.310 ± 2.025, bonding: 6.970 ± 1.855) but not intra-status cohesion (control: 6.220 ± 1.870, bonding: 6.240 ± 2.035). Data are plotted as box plots for each condition, with horizontal lines indicating median values, boxes indicating 25% and 75% quartiles, and whiskers indicating the 2.5%–97.5% percentile range. Cross symbols in each box represent the mean values. Data points outside the range are shown separately as circles. *$p < 0.05$, **$p < 0.01$, ***$p < 0.001$. Data used to generate Fig 1D–1H can be found in S1 Data. fNIRS, functional near-infrared spectroscopy; rDLPFC, right dorsolateral prefrontal cortex; rTPJ, right temporal-parental junction.

bonding ($F_{1, 173} = 46.570$, $p = 1.429 \times 10^{-10}$, $\eta^2 = 0.212$, ANCOVA, controlling for the total length of utterances, S2 Fig), suggesting that group members communicated more often under in-group social bonding. Moreover, social bonding increased leader's utterance to a greater degree than followers (hierarchy × bonding: $F_{1, 173} = 3.977$, $p = 0.048$, $\eta^2 = 0.022$; leader: $t_{174} = 5.427$, $p = 1.897 \times 10^{-7}$, Cohen's $d = 0.818$, 95% CI: 2.651, 5.682; follower: $t_{174} = 4.273$, $p = 3.172 \times 10^{-5}$, Cohen's $d = 0.644$, 95% CI: 1.389, 3.767, S2 Fig).

The number of turn-transition and turn-response time was then calculated separately for inter-status (a discrete pair of utterances between a group leader and a follower) and intra-status (a discrete pair of utterances between 2 followers) communications. We compared these measurements between social bonding and control conditions using hierarchy (leader versus follower) × bonding (bonding versus control) ANCOVAs (controlling for the total length of utterances) and corresponding linear mixed models (LMMs, with hierarchy and bonding as fixed effects and group as a random effect). This analysis revealed that social bonding increased the frequency of inter-status (compared to intra-status), communications to a great extent (increased the number of leader–follower turn transitions, bonding × hierarchy: $F_{1, 172} = 9.951$, $p = 0.002$, $\eta^2 = 0.055$, inter-status: $t_{174} = 5.658$, $p = 6.182 \times 10^{-8}$, Cohen's $d = 0.853$, 95% CI: 1.885, 3.904, intra-status: $t_{173} = 1.587$, $p = 0.114$, Cohen's $d = 0.240$, 95% CI: −0.206, 1.900; LMM: $F_{1, 347} = 7.673$, $p = 0.006$, Fig 1D) and shortened the turn response time (bonding main effect: $F_{1, 166} = 9.793$, $p = 0.002$, $\eta^2 = 0.056$, especially for inter-status turns: $t_{172} = -3.406$, $p = 0.001$, Cohen's $d = -0.516$, 95% CI: −0.154, −0.040; LMM: $F_{1, 340} = 12.924$, $p = 3.72 \times 10^{-4}$, Fig 1E). These results together suggested that social bonding was efficient in increasing group communication, especially promoted more frequent and responsive inter-status interactions and strengthened inter-status social connections.

At the end of the experiment, we asked participants to report subjective evaluations on inter- and intra-status cohesion. First, we found that the frequency of group communication predicted group cohesion. In groups with more frequent communications, their group members reported a higher level of group cohesion ($r_{172} = 0.206$, $p = 0.006$). Interestingly, more inter-status turn transitions selectively predicted inter-status cohesion ($r_{173} = 0.165$, $p = 0.029$, Fig 1F), but not intra-status cohesion ($r_{173} = 0.052$, $p = 0.496$). Similarly, more intra-status turn transitions predicted a higher level of intra-status cohesion ($r_{172} = 0.162$, $p = 0.033$, Fig 1G, but not inter-status cohesion, $r_{172} = 0.146$, $p = 0.055$). Second, social bonding selectively facilitated inter-status cohesion ($t_{174} = 2.261$, $p = 0.025$, Cohen's $d = 0.340$, 95% CI: 0.082, 1.238) rather than intra-status cohesion ($t_{174} = 0.040$, $p = 0.968$, Cohen's $d = 0.010$, 95% CI: −0.562, 0.602), confirmed by a significant interaction between hierarchy (inter- versus intra-status) and bonding (bonding versus control) on in-group cohesion rating ($F_{1, 174} = 4.914$, $p = 0.028$, $\eta^2 = 0.027$, Fig 1H). In addition, within-group interactions under social bonding were also perceived as more frequent and cohesive by third-party observers (Methods, S3 Fig).

## In-group social bonding influenced leader behavior and social perception of leader

Next, we examined whether social bonding influenced behaviors toward in- and out-group members differently (or not) in individuals of different social statuses (i.e., group leader and followers). Participants completed 2 economic games related to intergroup discrimination: (i) an intergroup dictator game (IDG) where participants donated to in-group and out-group members [15,58]; (ii) an intergroup prisoner's dilemma-maximizing differences game (IPD-MDG) where participants self-sacrificed separately to benefit in-group members ("ingroup love") and to derogate out-group members ("outgroup hate") [59,60]. We found

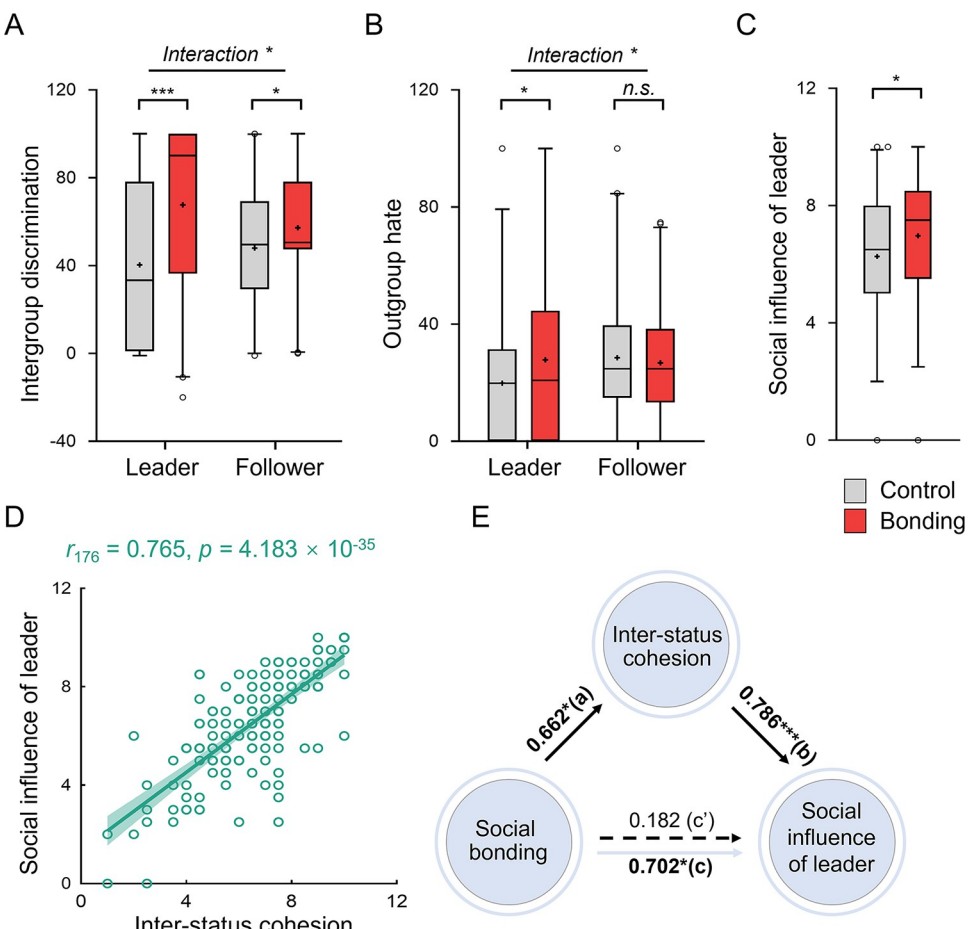

**Fig 2. The effect of in-group social bonding on leader behavior and the perception of leader.** (**A**) In-group social bonding increased intergroup discrimination to a greater degree in leaders (control: 40.314 ± 39.547, bonding: 67.649 ± 38.752) than followers (control: 47.971 ± 28.122, bonding: 57.261 ± 26.918). (**B**) Bonding increased out-group hate in leaders (control: 19.858 ± 22.151, bonding: 27.814 ± 29.810) but not in followers (control: 28.557 ± 20.210, bonding: 26.810 ± 18.040). (**C**) Under social bonding, followers perceived greater social influence of the leader (control: 6.270 ± 2.061, bonding: 6.970 ± 1.963). Data are plotted as box plots for each condition, with horizontal lines indicating median values, boxes indicating 25% and 75% quartiles and whiskers indicating the 2.5%–97.5% percentile range. Cross symbols in each box represent the mean values. Data points outside the range are shown separately as circles. (**D**) Leader's social influence was positively associated with inter-status cohesion (Pearson's correlation analysis). Each solid line represents the least squares fit, with shading showing the 95% *CI*. (**E**) Bonding increased perceived social influence of the leader through enhancing inter-status cohesion. $^*p < 0.05$, $^{***}p < 0.001$. Data used to generate Fig 2A–2E can be found in S1 Data.

that groups in the bonding (versus control) condition donated more to in-group members than to out-group members in the IDG ($F_{1,\ 174} = 26.406$, $p = 7.375 \times 10^{-7}$, $\eta^2 = 0.132$, Fig 2A), and such bonding-facilitated intergroup discrimination was stronger in group leaders than followers (hierarchy × bonding: $F_{1,\ 174} = 6.109$, $p = 0.014$, $\eta^2 = 0.034$; leader: $t_{174} = 4.631$, $p = 7.087 \times 10^{-6}$, Cohen's $d = 0.698$, 95% *CI*: 15.685, 38.983; follower: $t_{174} = 2.239$, $p = 0.026$, Cohen's $d = 0.338$, 95% *CI*: 1.101, 17.479; LMM: $F_{1,\ 348} = 6.256$, $p = 0.013$, Fig 2A). In the IPD-MDG, participants showed stronger in-group love (paired $t$ test: ingroup love (Mean ± SD): 34.332 ± 16.650, outgroup hate: 26.319 ± 15.764, $t_{175} = 3.966$, $p = 1.065 \times 10^{-4}$, Cohen's $d = 0.299$, 95% *CI*: 4.025, 12.001). Interestingly, the interactive effect of bonding and hierarchy was observed in out-group hate ($F_{1,\ 172} = 4.470$, $p = 0.036$, $\eta^2 = 0.025$; leader: $t_{172} = 1.995$, $p = 0.048$, Cohen's $d = 0.302$, 95% *CI*: 0.084, 15.830; follower: $t_{174} = -0.605$, $p = 0.546$,

Cohen's $d = -0.091$, 95% $CI$: $-7.443$, $3.950$; LMM: $F_{1, 346} = 3.898$, $p = 0.049$, Fig 2B) but not in in-group love ($F_{1, 172} = 0.445$, $p = 0.506$, $\eta^2 = 0.003$). Taken together, in-group social bonding increased intergroup discrimination and "hate" towards outgroup, especially in group leaders.

We concluded our behavioral analysis by examining how in-group social bonding influenced followers' perception of the leader (i.e., leaders' social influence and social attraction). We found that followers in groups under social bonding (versus control) perceived their leaders as more influential ($t_{174} = 2.313$, $p = 0.022$, Cohen's $d = 0.348$, 95% $CI$: $0.103$, $1.301$, Fig 2C) and more attractive ($t_{174} = 2.944$, $p = 0.004$, Cohen's $d = 0.444$, 95% $CI$: $0.265$, $1.339$, S4A Fig). Moreover, the perceived social influence and social attraction of leaders were positively associated with in-group cohesion, especially evident with inter-status cohesion (social influence: $r_{176} = 0.765$, $p = 4.183 \times 10^{-35}$, Fig 2D; social attraction: $r_{176} = 0.702$, $p = 1.743 \times 10^{-27}$, S4B Fig; weaker but also with intra-status cohesion, social influence: $r_{176} = 0.435$, $p = 1.548 \times 10^{-9}$, S4C Fig; attraction: $r_{176} = 0.287$, $p = 1.107 \times 10^{-4}$, S4D Fig; slope test: social influence: z = 5.04, $p = 1.164 \times 10^{-7}$; attraction: z = 5.36, $p = 2.081 \times 10^{-8}$), suggesting that followers perceived their leaders as more influential and attractive in more cohesive groups, especially when they coordinated better with the leaders. Importantly, we established a potential mediation path that the effects of social bonding on perceived social influence (Indirect effect = 0.520, $SE = 0.238$, 95% bootstrap $CI$: $0.073$, $1.001$, Sobel test, $Z = 2.224$, $p = 0.025$, Fig 2E) and attraction (Indirect effect = 0.426, $SE = 0.196$, 95% bootstrap $CI$: $0.057$, $0.0832$, Sobel test, $Z = 2.224$, $p = 0.026$, S4E Fig) in leaders were fully mediated by inter-status cohesion (Methods).

The behavioral results together suggested that social bonding (i) increased contributions of group leader in both within-group communication and intergroup conflict; and (ii) selectively strengthened inter-status (but not intra-status) communication and cohesion, which possibly resulted in a better impression and more social influence of group leader.

## In-group social bonding selectively increases inter-status neural synchronization in the rDLPFC

We applied fNIRS to each hierarchical group and simultaneously recorded all group members' neural activity, captured by the dynamic hemodynamic signals, from the rDLPFC (7 channels, Fig 1B) and the right temporoparietal junction (rTPJ, 7 channels, Fig 1B), during resting-state and interaction stages. Consistent with previous studies [15,34–37], we operationalized the INS in terms of wavelet transform coherence (WTC). The WTC value indicates the cross-correlation between 2 fNIRS time series of concentration changes in oxygenated hemoglobin (oxy-Hb) in dyads of individuals as a function of frequency and time. Within each three-person group, we calculated the coherence values from the leader–follower dyads to index the inter-status INS, and the coherence value from the follower–follower dyads to index the intra-status INS (Methods, Fig 3A). We were interested in the INS specific to group interaction, thus focused on the INS increases during group interaction relative to the resting-state. We compared coherence values between the resting-state and group interaction to identify the frequency band of interest (FOI, Methods, Fig 3A). Moreover, the INS specific to group interaction was indicated by the FOI-averaged coherence differences (Group interaction—Resting) and then submitted into the following analyses.

To examine whether social bonding differently influenced the inter- and intra-status INS, we submitted INS in each channel of the rDLPFC and rTPJ to 2 (hierarchy: inter- versus intra-status dyads) × 2 (bonding: bonding versus control) mixed-model ANOVAs and LMMs (hierarchy and bonding as fixed effects, group as a random effect). Significant effects were identified after false discovery rate (FDR)-corrected for multiple comparisons for 14 channels. First, the analysis revealed stronger INS in the rTPJ for the inter-status than intra-status dyads (main

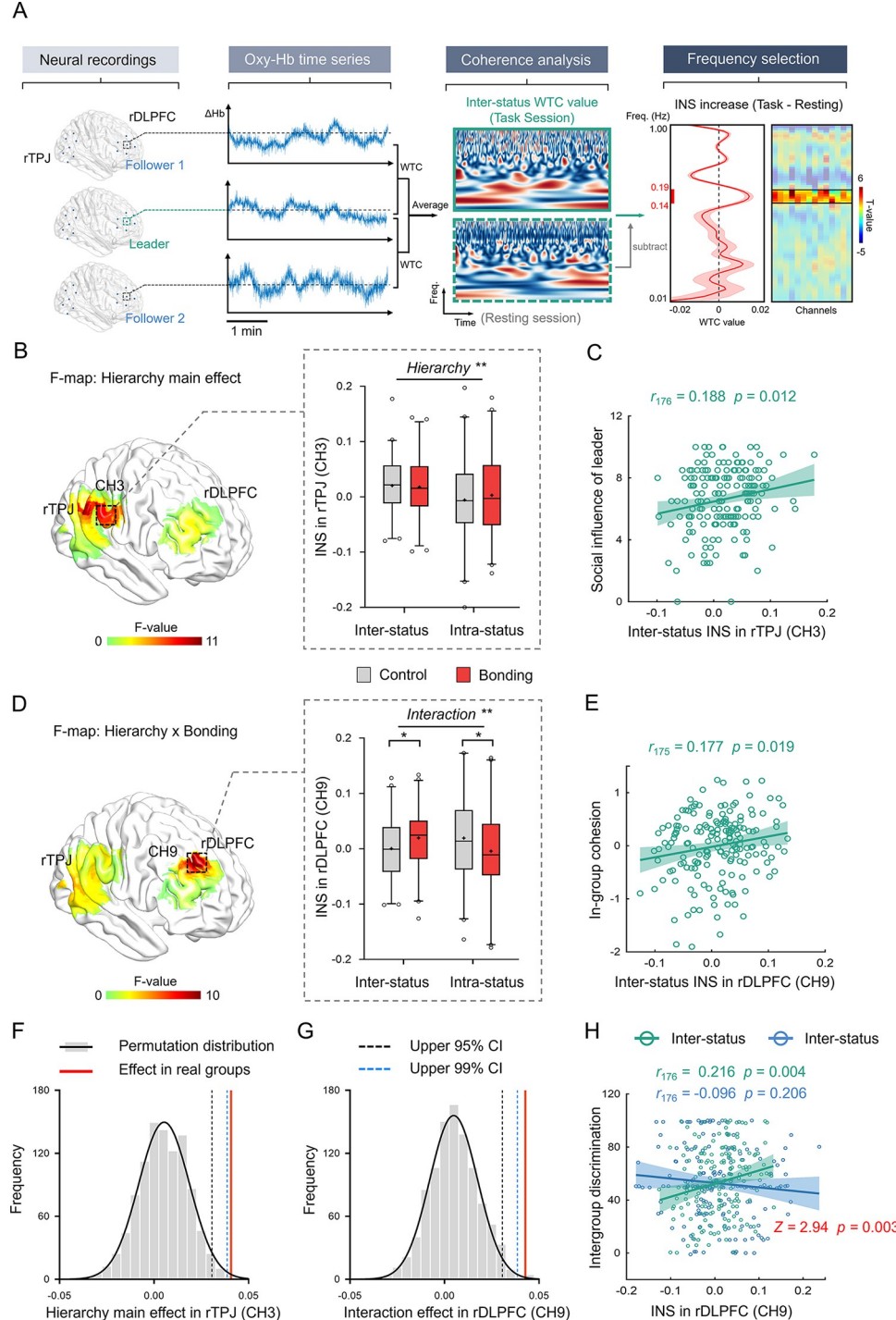

**Fig 3. In-group social bonding selectively increases inter-status neural synchronization in the rDLPFC. (A)**
Illustration of inter-status neural synchronization calculation. The concentration changes in oxygenated hemoglobin
(oxy-Hb) were simultaneously collected in each channel from each member of the three-person group. The cross-
correlations between oxy-Hb time series of leader–follower pairs were generated through WTC analysis, and the 2
pairs were then averaged to indicate INS of inter-status dyads. Comparison of coherence values between group
interaction and resting-state identified INS specific to group interaction and the frequency band of interest. (**B**)
Stronger inter-status (vs. intra-status) INS at channel 3 in the rTPJ. Data are plotted as box plots for each condition,
with horizontal lines indicating median values, boxes indicating 25% and 75% quartiles and whiskers indicating the
2.5%–97.5% percentile range. Cross symbols in each box represent the mean values. Data points outside the range are

shown separately as circles. (**C**) Positive association between inter-status INS at channel 3 in the rTPJ and group leader's social influence (Pearson's correlation analysis). The solid line represents the least squares fit, with shading showing the 95% *CI*. (**D**) Bonding increased inter-status INS at channel 9 in the rDLPFC (control: 0.001 ± 0.053, bonding: 0.019 ± 0.055) but decreased that of intra-status dyads (control: 0.019 ± 0.076, bonding: −0.004 ± 0.079). (**E**) Positive association between inter-status INS at channel 9 in the rDLPFC and perceived group cohesion. (**F/G**) INS validation by nonparametric permutation tests. We compared the hierarchy main effect in the rTPJ and the interaction effect in the rDLPFC of real group against within-condition permutation distributions (*n* = 1,000). The observed effects of hierarchy in the rTPJ (**F**) and of hierarchy × bonding interaction in the rDLPFC (**G**) exceeded the upper limits of 99% *CI* of the permutation distributions. (**H**) The inter-status (but not intra-status) INS at channel 9 in the rDLPFC was associated with intergroup discrimination. *$p$ < 0.05, **$p$ < 0.01. Data used to generate Fig 3B–3H can be found in S1 Data. INS, inter-brain neural synchronization; rDLPFC, right dorsolateral prefrontal cortex; rTPJ, right temporal-parental junction; WTC, wavelet transform coherence.

effect of hierarchy at channel 3, $F_{1, 174}$ = 10.207, $p$ = 0.002, $\eta^2$ = 0.055, survived 14-channel-wise FDR-correction; LMM: $F_{1, 348}$ = 9.684, $p$ = 0.002, Fig 3B and S2 Table). Moreover, stronger inter-status INS in the rTPJ was associated with stronger social influence of the group leader ($r_{176}$ = 0.188, $p$ = 0.012, Fig 3C), suggesting that followers perceived their leaders more influential when their rTPJ activity synchronized with that of leader to a greater degree.

Interestingly, we observed a significant hierarchy × bonding interaction on INS at channel 9 in the rDLPFC ($F_{1, 174}$ = 9.577, $p$ = 0.002, $\eta^2$ = 0.052, survived 14-channel-wise FDR-correction; LMM: $F_{1, 348}$ = 8.912, $p$ = 0.003, Fig 3D and S2 Table). Specifically, social bonding selectively increased the inter-status INS (independent-sample *t* tests, $t_{174}$ = 2.357, $p$ = 0.020, Cohen's *d* = 0.355, 95% *CI*: 0.003, 0.035) but decreased intra-status INS ($t_{174}$ = −1.998, $p$ = 0.047, Cohen's *d* = −0.301, 95% *CI*: −0.047, 0.000). Taking another perspective to interpret the interaction, we observed stronger inter-status (versus intra-status) INS in the bonding condition (paired-sample *t* tests, $t_{88}$ = 2.521, $p$ = 0.013, Cohen's *d* = 0.267, 95% *CI*: 0.005, 0.043), which was comparable even in an opposite trend in the control condition ($t_{86}$ = −1.875, $p$ = 0.064, Cohen's *d* = −0.201, 95% *CI*: −0.039, 0.001). Interestingly, we found that inter-status but not intra-status INS in the rDLPFC was predictive of how the group interaction was perceived. Independent rater perceived the groups with stronger inter-status INS in rDLPFC more cohesive (inter-status INS: $r_{175}$ = 0.177, $p$ = 0.019, Fig 3E; intra-status INS: $r_{175}$ = 0.085, $p$ = 0.265).

We next conducted 2 sets of validation analyses to exclude the possibility that the observed INS was partially reflected participants sharing the same environment or performing the same task. First, within the bonding and control conditions, we generated 176 within-condition three-person pseudo-groups by randomly grouping a real leader and 2 real followers from different original groups in the same bonding or control condition as 1 three-person pseudo-group (Methods, S5A Fig). We recalculated the inter- and intra-status INS for pseudo groups and repeated these procedures for 1,000 times. We then conducted nonparametric permutation tests on the observed effects of the real interacting groups against the 1,000 permutation samples. This analysis confirmed that both the main effect of hierarchy in the rTPJ (real group: 0.041, permutation: 95% CI: −0.020, 0.031, 99% CI: −0.029, 0.039, $p$ = 0.003, Fig 3F) and the interactive effect of hierarchy × bonding in the rDLPFC (real group: 0.043, permutation: 95% CI: −0.020, 0.031, 99% CI: −0.028, 0.039, $p$ = 0.003, Fig 3G) in the real groups exceeded the upper limit of 99% *CI* of the permutation distributions. Second, similar analysis conducted on the cross-condition permutation samples (i.e., 1 leader and 2 followers randomly from the bonding or control condition were organized into a pseudo-group, S5B Fig) again confirmed the observed effects in the real interacting groups (S5C and S5D Fig). Taken together, the 2 validation analyses confirmed that the observed bonding and/or hierarchy effects on INS in the real interactive groups were not due to same experimental environment or performing the same task. In addition, we further eliminated potential influence of global physiological noises

by (i) using a wavelet-based denoising method [61]; and (ii) controlling the globally co-varying signals in the hierarchy × bonding ANCOVAs (i.e., including the global mean of INS across all channels as a covariant, [62]). These 2 complementary analyses well replicated the aforementioned patterns (S3 Table).

## In-group social bonding influences group behavior via inter-status INS in the rDLPFC

Next, we aimed to reveal whether and how the inter-status or intra-status INS within a group was linked to behaviors towards in-group and out-group members. We found that stronger inter-status (but not intra-status) INS in the rDLPFC was predictive of stronger intergroup discrimination (i.e., donations to in-group versus out-group members; inter-status: $r_{176} = 0.216$, $p = 0.004$; intra-status: $r_{176} = -0.096$, $p = 0.206$, Fig 3H). Further modulation analysis compared the Fisher-transformed correlation coefficients and confirmed selective prediction of inter- (versus intra-) status INS on intergroup discrimination ($z = 2.94$, $p = 0.003$). These results suggested that leader and followers synchronized their rDLPFC activity in a way that predicted how they differently treated in-group and out-group members.

## In-group social bonding facilitates leader-to-follower neural alignment in the rDLPFC

Next, we aimed to probe the directionality of the inter-status neural synchronization. We specifically asked whether social bonding influenced the leader-to-follower (i.e., neural activity of leaders preceded that of followers) or follower-to-leader (i.e., neural activity of followers preceded that of leaders) neural alignment, or both. The leader-to-follower neural alignment reflects situations in which the group leader leads and followers follow, while the follower-to-leader neural alignment may indicate instances where followers take the lead and group leader follows. We thus conducted time-lag analysis that has been employed in previous studies to reveal the directional influence between leader and follower's neural activity [63–65]. The time course of leader's neural activity was shifted relative to that of the followers from −10 to 10 s (in 1-s increment). Positive time lags indicated leader-to-follower neural alignment and negative time lags reflected follower-to-leader neural alignment (Fig 4A). On each time lag, the coherence values of inter-/intra-status dyads were recomputed for both resting-state and interaction stage. The coherence value increase (i.e., lagged INS during interaction minus that during resting) were used to indicate lagged neural alignment and submitted into subsequent analysis (Methods). It should be noted that the time lag analyses were conducted for the channels that showed increased neural synchronization between leader and followers during the interaction stage, i.e., channel 3 in the rTPJ and channel 9 in the rDLPFC.

While the time-lagged neural alignment in the rTPJ was significant in both the leader-to-follower and follower-to-leader directions (with peak centered at 0 second, S6A Fig), regardless of the bonding/control conditions (S6B Fig), the time-lagged neural alignment in the rDLPFC was significant mainly in the leader-to-follower direction and modulated by social bonding. We conducted independent $t$ tests on time-lagged neural alignment between bonding and control conditions on each time lag and revealed a right-skewed bell-shaped curve for the bonding effect (Fig 4B). To be specific, bonding (relative to control condition) significantly facilitated inter-status neural alignment on +1 to +6 time lags (with peak centered at +5 s, Fig 4B and S4 Table), indicating that leaders' neural activity preceded that of followers for 1 to 6 s (survived multiple corrections for 21 time lags). Separate analyses for bonding and control conditions confirmed significant increase (contrast to zero) of inter-status neural alignment only occurred in the bonding condition (Fig 4C and S5 Table); however, neither the leader-to-

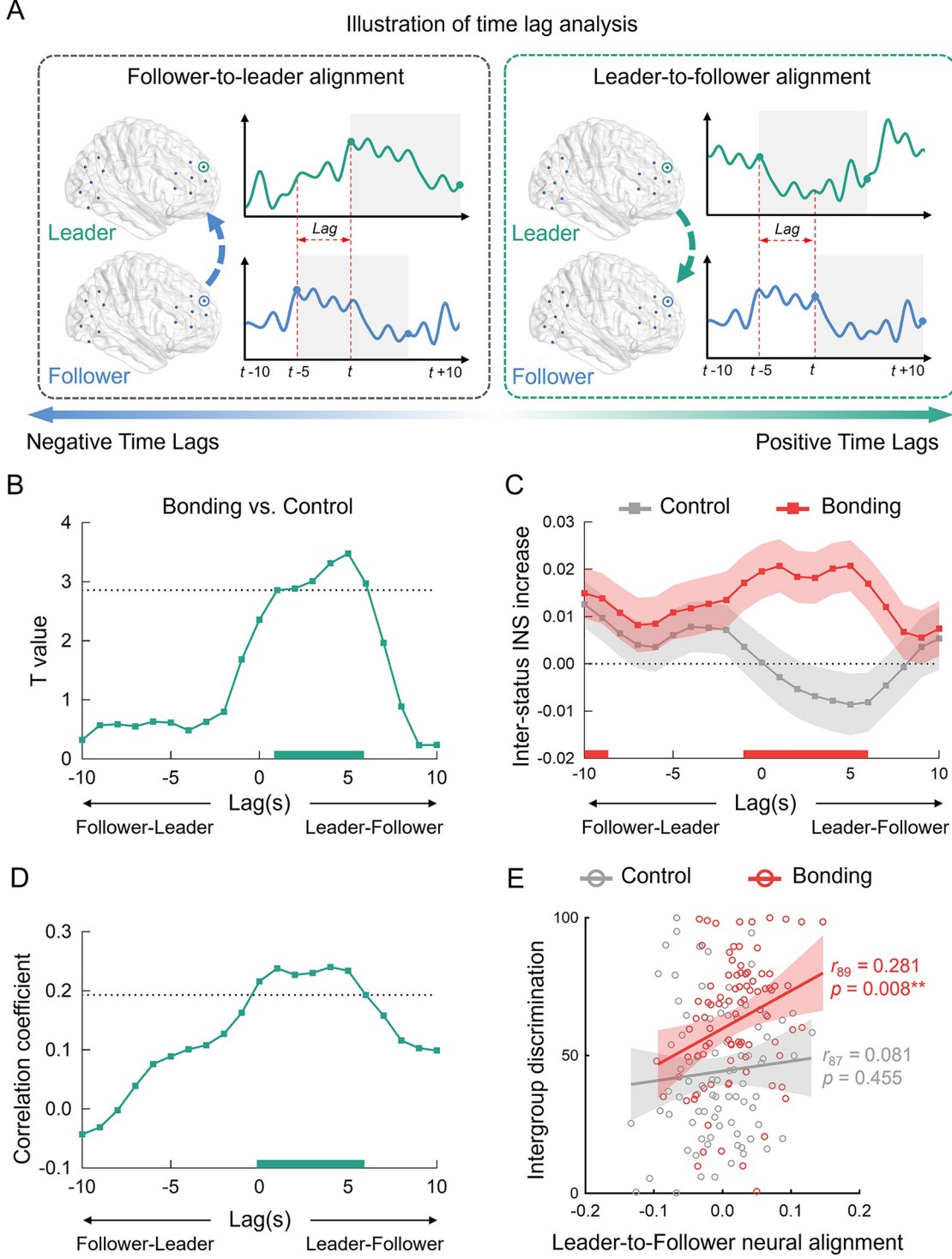

**Fig 4. In-group social bonding facilitates leader-to-follower neural alignment in the rDLPFC.** (**A**) Illustration of time lag analysis. The neural activity of the leader is shifted forwards (backwards) in relative to that of followers in the positive (negative) time lags, indicating leader-to-follower (follower-to-leader) alignment. (**B**) Bonding (vs. control) facilitated leader-to-follower neural alignment on +1 to +6 time lags (peaked at +5 s), survived FDR multiple correction. The dashed line indicates the corrected significance threshold. The significant time lags (survived multiple correction) are highlighted with the horizontal line on the x-axis. (**C**)

Significant increases of inter-status neural alignment on +1 to +6 time lags was only found in the bonding condition. Shaded areas represent SE. (**D/E**) Inter-status neural alignment was positively correlated with intergroup discrimination on +1 to +6 time lags in the bonding (but not control) conditions (correlation coefficients on each time lag of −10 to +10 lags, **D**; correlations for the averaged inter-status neural alignment, **E**). Correlations were performed by Pearson's correlation coefficient analysis. Each solid line represents the least squares fit, with shading showing the 95% *CI*. Data used to generate Fig 4B–4E can be found in S1 Data. FDR, false discovery rate; rDLPFC, right dorsolateral prefrontal cortex; SE, standard error.

follower nor follower-to-leader neural alignment was significant in the control condition (Fig 4C).

Moreover, such leader-to-follower neural alignment facilitated intergroup discrimination. Specifically, we correlated leader-to-follower neural alignment with intergroup discrimination and showed that stronger leader-to-follower neural alignment (when leaders' neural activity preceded that of followers +1 to +6 s) predicted larger intergroup discrimination (Fig 4D, FDR corrected for 21 time lags, S6 Table). Moreover, this relationship was especially true in the bonding but not control condition (averaged neural alignment of +1 to +6 time lag, bonding: $r_{89} = 0.281$, $p = 0.008$, control: $r_{87} = 0.081$, $p = 0.455$, Figs 4E and S7 for each time lag separately). Control analyses were conducted for the intra-status INS for −10 to 10 s (in 1-s increment). Neither the bonding effect (S8A Fig) nor the correlation with intergroup discrimination (S8B Fig) was significant on intra-status neural alignment at any time lags. Taken together, the bonding-elevated inter-status INS selectively emerged when leaders' neural activity preceded that of followers, indicating that leaders predicted or anticipated followers' mental states and followers tracked the leader's mental states to achieve the leader-to-follower neural alignment under in-group social bonding condition. Such leader-to-follower neural alignment may further result in stronger intergroup discrimination under in-group social bonding.

## Stronger rDLPFC-rTPJ functional connectivity in the leader accounted for leader-to-follower neural alignment

The observation that the bonding effect was selectively exhibited on the inter-status dyads, especially in a leader-to-follower manner, led us to further examine the bonding effects respectively in leaders and followers. The functional connectivity between rDLPFC and rTPJ has been shown to play a key role in perspective taking, mental inference, and information integrating [66,67]. As leader-to-follower neural alignment may indicate situations in which group leaders predict followers' mental states or perspectives [64], as well as when followers strategically attend to and track the group leader [68], we compared rDLPFC-rTPJ functional connectivity between leaders than followers. Furthermore, we tested whether rDLPFC-rTPJ functional connectivity could account for the leader-to-follower neural alignment. To this end, we applied cross-correlation analysis to assess the functional connectivity of rDLPFC and rTPJ in leaders and followers (Methods). Results showed that, leaders (versus followers) showed stronger rDLPFC-rTPJ connectivity (Fig 5A for channel-pairwise rDLPFC-rTPJ connectivity, 28 rDLPFC-rTPJ channel pairs survived FDR correction for 49 channel pairs, S7 Table; Fig 5B for the grand mean rDLPFC-rTPJ connectivity, two-way mixed-model ANOVA, $F_{1, 174} = 12.006$, $p = 6.679 \times 10^{-4}$, $\eta^2 = 0.065$; LMM: $F_{1, 348} = 12.438$, $p = 4.77 \times 10^{-4}$). We next correlated the strength of rDLPFC-rTPJ connectivity (averaged connectivity between channel 9 in the rDLPFC and each channel in the rTPJ) in leaders and followers respectively with the leader-to-follower neural alignment at channel 9 in the rDLPFC (Methods). Results endorsed a positive relationship between rDLPFC-rTPJ functional connectivity in leaders (but not followers) and the leader-to-follower neural alignment (leader: $r_{176} = 0.177$, $p = 0.019$, Fig 5C, for average of lagged inter-status neural alignment, S9 Fig for correlations to neural alignment on each time

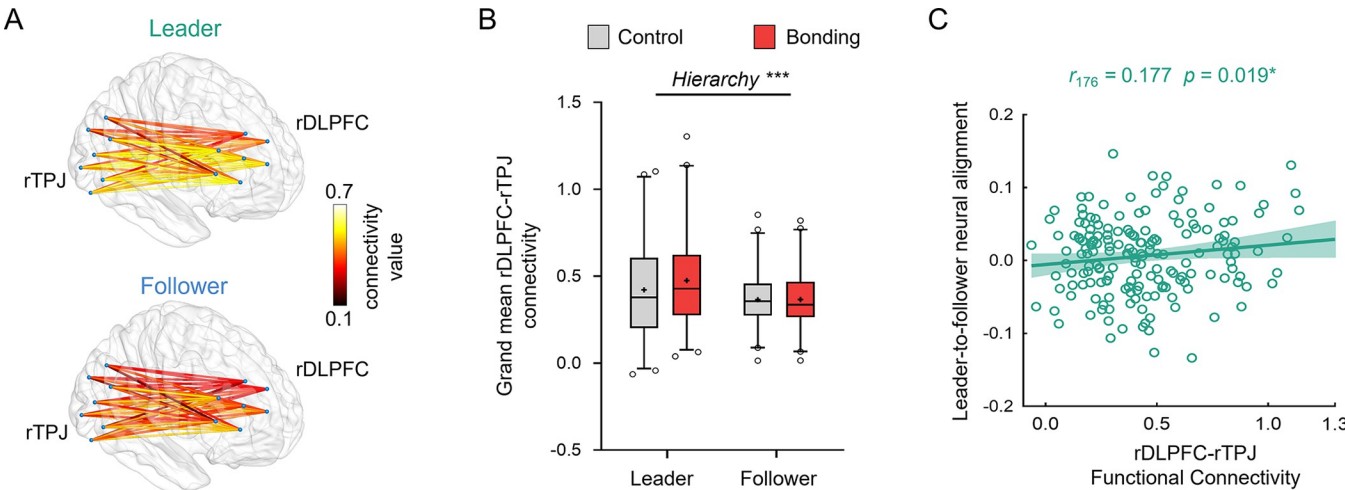

**Fig 5. Stronger rDLPFC-rTPJ functional connectivity in leaders accounted for leader-to-follower neural alignment.** (A/B) Leaders (vs. followers) showed stronger rDLPFC-rTPJ functional connectivity at channel-pairwise level (28 rDLPFC-rTPJ channel pairs survived FDR correction for 49 channel pairs, **A**) and the grand mean level (i.e., averaged coherence value of 49 channel pairs between the rDLPFC and rTPJ, **B**, displayed as box plots, with horizontal lines indicating median values, boxes indicating 25% and 75% quartiles and whiskers indicating the 2.5%–97.5% percentile range. Cross symbols in each box represent the mean values. Data points outside the range are shown separately as circles). (**C**) The functional connectivity between channel 9 in the rDLPFC and the rTPJ (averaged across 7 channel pairs) was associated with leader-to-follower neural alignment at channel 9. The solid line represents the least squares fit, with shading showing the 95% *CI*., **\*\****p* < 0.01. Data used to generate Fig 5A–5C can be found in S1 Data. FDR, false discovery rate; rDLPFC, right dorsolateral prefrontal cortex; rTPJ, right temporal-parental junction.

lag; follower: $r_{176} = 0.004$, $p = 0.955$), suggesting that leaders with stronger rDLPFC-rTPJ connectivity predicted their followers' neural activity to a greater degree.

## Discussion

Social bonding has been recognized as a potent strategy to enhance in-group interaction and cohesion across species [16–21] and to reinforce hierarchical structures [22]. However, to date, the neurocognitive mechanisms underlying the effects of social bonding on hierarchical interactions remain largely elusive. We applied a multi-brain hyper-scanning approach to real-time within-group communication and differentiated interpersonal interactions within a hierarchical group into 2 status-related types: inter-status and intra-status interactions. By doing so, we provide evidence that in-group social bonding exerts distinct effects on individuals of different statuses and on dyads of different status-related relationships. Specifically, social bonding selectively facilitates communication frequency and responsiveness, as well as synchronized and leader-proceeding neural alignment for inter-status dyads but not for dyads with the same social status. These findings distinguish between inter-status and intra-status interactions and shed lights on the distinct neurocognitive mechanisms through which social bonding shapes group dynamics in hierarchical groups.

Hierarchical groups are characterized by complex relationships among group members [3], where their behaviors and neural systems dynamically coordinate and interact [4]. However, most neuroscience studies on leadership and social hierarchy have examined the brains of leaders or followers in isolation [69,70], focusing on leading behaviors or the distinct roles leaders and followers respectively played in a group [71]. This approach has overlooked the interactive nature of hierarchical relationships, limiting our understanding of how different individuals dynamically interact within a hierarchical group. Recently, the emergence of second-person, interactive neuroscience has underscored the significance of studying how the brains of socially interacting individuals entrain to support social interaction and relationships

[31–33]. Through the lens of interactive neuroscience, we were able to characterize distinct behavioral and neural profiles of 2 types of interpersonal interactions based on the status relationship of interactive dyads. Within a hierarchical group, group members of different, rather than same, status engaged in more frequent and rapid information exchange, and frequent inter-status interaction was closely related to in-group cohesion for the aggregated group. At the neural level, inter-status interaction was featured with stronger neural synchronization in the rTPJ, a key brain region involved in mentalizing and taking perspectives of others, especially dissimilar others [72]. This finding suggests an important role of perspective-taking and social mentalizing in inter-status interaction, while interactions among individuals with the same status may require less mentalizing efforts and gain social support through "shared mind." Together, the distinct behavioral and neural profiles for inter-status and intra-status interactions highlight the importance of differentiating status-related interactions and relationships in future work to understand leadership and group dynamics in hierarchical groups.

By differentiating between inter-status and intra-status interactions in hierarchical groups, we are able to arbitrate between 2 possible mechanisms: whether social bonding enhances group dynamics in hierarchical groups regardless of the interaction types (i.e., enhancing both inter-status and intra-status interactions in a similar way), or if it is modulated by the interaction type, selectively promoting either inter- or intra-status interaction. Our results, which revealed distinct behavioral and neural effects of social bonding on inter- and intra-status dyads, provide evidence in support of the latter. Specifically, social bonding mainly facilitates inter-status (rather than intra-status) communication and cohesion, and selectively enhances neural alignment between the rDLPFC activity of leaders and followers. Synthesizing and extending findings from previous studies [13,15], we suggest that the mechanisms by which social bonding operates within a group depends on the type of group structure. Unlike non-hierarchical groups in which social bonding fosters interaction and cohesion among group members indiscriminately, social bonding in hierarchical groups primarily serves to foster and reinforce connections among individuals with different social statuses. It selectively enhances mutual understanding and information exchange between leaders and followers rather than among peers, highlighting the importance of status-related interactions in shaping the effects of social bonding within a hierarchical group.

We further investigated how social bonding facilitated neural entrainment between individuals of high and low status, specifically addressing whether this entrainment occurred in a high-to-low status or low-to-high status direction or both (i.e., a bidirectional manner). By employing time-lagged analysis, we revealed the temporal dynamics of the inter-status synchronization. The prefrontal activity of the group leader preceded that of followers by 1 to 6 s, indicating a unidirectional neural alignment from group leader to followers after social bonding. Previous studies have associated such sequential neural alignment with anticipatory processing, reflecting active engagement and predictive processing of others' behaviors and intentions during social interactions [63–65]. For example, in dyadic communication, the listener's prefrontal activity often precedes that of the speaker, and such anticipatory neural response facilitated mutual understanding and successful communication [65]. Therefore, our findings of leader-to-follower neural alignment suggested a potential underlying process through which social bonding influenced inter-status interaction: the group leader actively engaged in anticipating and predicting the mental states of followers, enabling more frequent communication and coordination with them. Moreover, the effects of social bonding on synchronized and leader-proceeding neural entrainment were only observed in the rDLPFC rather than the rTPJ. Previous work has evidenced a crucial role of the prefrontal cortex, particularly the DLFPC, in predicting forthcoming sensory inputs (e.g., short sound, [73]), social cues (e.g., eye gaze, [63]), and social dominance and future social interactions [74], suggesting

DLPFC as a hub region for monitoring errors and updating predictions of upcoming inputs to prepare for appropriate responses [73]. Therefore, the impact of social bonding on rDLPFC synchronization and leader-to-follower alignment may provide a neurocognitive account for increased leader initiation and more frequent, efficient leader-follower information exchange. Interestingly, we showed that group leaders with stronger DLPFC-TPJ functional connectivity exhibited a greater degree of neural alignment with followers. This link suggested that the exchange and integration of information between DLPFC and TPJ may support predictive neural alignment from leader to follower. Taken together, in-group social bonding may enable the group leader to actively adopt followers' perspective, consider their potential behaviors and intentions, and align predictively with them.

In-group social bonding prioritizes the allocation of cognitive resources, emotional attachments, and neural entrainment to inter-status interactions within hierarchical groups, and fails to yield comparable effects on intra-status interactions. This discrepancy cannot be attributed to a ceiling effect resulting from potentially preexisting close bonding among individuals of the same-status due to shared similarities [75], which may limit the extent to which external bonding manipulation can further enhance their interactions. This interpretation is supported by the absence of differences in communication frequency and responsiveness, and neural synchronization between intra-status and inter-status dyads in the control condition without external bonding manipulation. Moreover, the bonding effects on neural synchronization exhibited a distinct pattern, with social bonding decreasing neural synchronization in the rDLPFC for intra-status dyads while increasing it for inter-status dyads. Given the critical role of the rDLPFC in top-down regulation of social attention [76], we propose that the observed decreases in neural synchronization in the rDLPFC of intra-status dyads may reflect a disengagement of attention from fellow members, potentially accompanied by a reallocation of attention towards the group leader [77]. Consistently, followers under social bonding engaged in more frequent and responsive communications with group leaders (compared to other fellow members, as shown in Fig 1E) and perceived leaders as more influential and socially attractive. Such an upward attention shift may contribute to the maintenance of structural stability by facilitating follower's understanding of the intentions and/or preferences of the group leader while minimizing potential competition and violations [74,77,78]. Together, these findings put forward the hypothesis that the effects of social bonding on modulating neural couplings and redirecting attentional engagement from intra-status to inter-status interaction may serve to reinforce the hierarchical structure.

Extending previous findings of bonding effects on egalitarian group, our work reveals the critical roles of social status in shaping the strength and nature of the social bonding experience in hierarchical groups, which operates at both the individual and dyadic levels. At the individual level, social bonding facilitates the initiation and engagement of high-status individuals in group communications, while increases the responsiveness of low-status individuals to group leader and their perceptions of high-status individuals. At the dyadic level, social bonding exerts distinct effects on the inter-status and intra-status dynamics, potentially through the mechanisms of top-down predictive alignment and bottom-up attentional shift. Specifically, social bonding increases a leader's forward-prediction of follower's neural activity, while suppresses follower's neural and attentional entrainment with same-status fellows. These plausible neurocognitive pathways helped us to synthesize social bonding effects, providing neural accounts for the effect of social bonding on group dynamics within hierarchical social contexts. The establishment of social bonds between leader and followers may serve to alleviate inter-status inequality and competition, foster inter-status coalitions, and maintain social hierarchy [71].

Our findings may be limited to the current experimental settings and could raise a number of exciting research questions for future studies. First, nonverbal communication, such as

gestures, facial expressions, and eye contact, plays a crucial role in real-life social interactions. However, since our participants were restricted to online communication through typing, the effects of social bonding on nonverbal hierarchical interaction remain unexplored. Second, the current study was conducted within simple three-person hierarchical groups with leaders who represented symbolic, perhaps prestige-style leaders, democratically elected by group members. These leaders made more contributions in intergroup economic games and established positive connections with followers, but lacked the authority to sanction fellow members or allocate resources. These settings deviated from those commonly encountered in complex real-life scenarios. Therefore, caution should be exercised when attempting to generalize our findings and future work is encouraged to explore the generality and specificity of social bonding effects across diverse leadership styles. Third, our study examined the three-person group, which is the minimal unit of a hierarchical group with only 2 levels of social hierarchy. While group dynamics and leadership in such a small-scale society are arguably representative, it lacks the complexity of group structure institutions. It will be interesting for future research to test whether the bonding effect would be weakened or amplified by the despotic power of non-human animal groups or the complexity of large-scale social networks.

Finally, it should be noted that the current dataset was obtained from participants of a specific cultural background, i.e., East Asian Chinese individuals. This raises the question of whether the observed effects in the current sample can be generalized to individuals from other cultures. Individuals from East Asian cultures place emphasis on group cohesion, interpersonal connection, and social hierarchy [79–81]. In comparison to Western cultures, followers in East Asian cultures tend to display higher levels of obedience and commitment towards their group leader while also encouraging more supportive leadership [82]. Therefore, one may expect cultural differences in how leaders and followers interact, especially after in-group social bonding, in hierarchical groups. To explore this possibility further, we conducted a preliminary examination by assessing individual differences in cultural values within our sample. Previous studies have suggested that cultural group differences in the neural activity underlying social cognition may be mediated by cultural values, such as interdependence of self-construal [83]. In the current study, we employed the Self-Construal Scale [84] to assess individual variations in cultural value of interdependence. We found that individual differences in independence did not influence behavioral and neural indices related to inter- or intra-status interactions nor did they affect observed bonding effects on behavioral and neural indices. These results suggested that our findings may be insensitive to culture-specific values. However, we acknowledge that the lack of modulation by cultural values could potentially be attributed to minimal variability in culture values within a single cultural context. It is important for future cross-cultural research to directly test whether and how our current findings can be generalized to other cultural populations.

## Materials and methods

### Participants

The current study reported behavioral and neural data from 176 three-person, same-gender groups (528 healthy volunteers). Eighty-nine three-person groups were randomly assigned to the bonding condition, and 87 groups underwent a matched no-bonding control condition. Participants in the bonding/control conditions, regardless of their role of leader or follower, did not differ significantly in age, gender, education, empathic capacity, cooperative personality traits, preference for social hierarchy, cultural value, or baseline intergroup discrimination (all $ps > 0.05$; S1 Table). The current study has not been preregistered. We conducted the study with a relatively large sample which was chosen to exceed comparable behavioral and

neuroimaging studies [85,86] to enhance power and to draw robust conclusions. Moreover, we conducted post hoc power analysis and confirmed that the bonding effects on both behavioral and neural indices yielded a statistical power larger than 99% (i.e., 99.8% for 1 main behavioral index of intergroup discrimination and 99.5% for 1 main neural index of the INS in the rDLPFC), based on the current sample ($n = 176$) with $\alpha = 0.05$, repeated measures $F$-tests, and effect size $f = 0.188$ (0.234). In addition, an independent group of participants ($n = 14$, 5 males, age: 18 to 23 years old, mean ± SD = 19.85 ± 1.61 years, education: 13 to 15 years, mean ± SD = 14.50 ± 1.09 years) were recruited to evaluate the group interaction quality for the 176 groups. All participants had a normal or corrected-to-normal vision and were free of psychiatric disorders and neurological conditions.

## Ethics statement

All participants provided written informed consent prior to the study and were paid for their participation. The experimental protocol was approved by the local research ethics committee at the State Key Laboratory of Cognitive Neuroscience and Learning, Beijing Normal University (protocol number: IORG0004944) and was conducted in accordance with the Declaration of Helsinki.

## Experimental procedure and tasks

Prior to the formal experiment, participants completed an online survey consisting of a set of questionnaires and color preference for black versus white (for the bonding manipulation). One to 4 days later, groups of 3 same-gender strangers came to the laboratory. During the fNIRS-based hyper-scanning, participants sat face-to-face in a triangle and completed 3 sessions (Figs 1A and S1): (i) a baseline session of 4-min resting-state where participants remained motionless, eyes closed, and mind relaxed [44]; (ii) a 4-min in-group social bonding (or no-bonding control) manipulation session; and (iii) a 4-min online within-group interaction session where participants discussed a given topic with in-group members and democratically elected the leader of their group. We employed online interaction in the current experiment as a growing number of social interactions are taking place online (e.g., video conference, [87]). In addition, the online setting ensures the anonymity of each group member, avoiding potential confounding. After completing the 3 sessions, participants were asked to report subjective evaluations on inter- and intra-status cohesion. Followers also reported perceived social influence and social attraction of their group leader. Moreover, participants completed an IDG and an IPD-MDG before the experiment and at the end of the experiment.

**In-group social bonding manipulation session.** Each three-person group was randomly assigned to the social bonding or no-bonding control session. The in-group bonding manipulation in the current study was adapted from several previously validated procedures [15,46–49]. Specifically, we merged 3 essential procedures used in previous studies to manipulate ingroup bonding: shared preference [46], symbolic marker [47], and similarity among group members [48]. First, for each in-group bonding session, we invited 3 participants preferring the same color (black or white, as indicated in the preexperiment survey) and used this color preference to create group identities, with the individuals referring black over white being labeled as "Group Black" and those preferring white over lack being labeled as "Group White." Second, each participant was then given either a black vest (for "Group Black") or a white vest (for "Group White") to wear during the entire experimental session. The same-colored uniforms serve as an arbitrary symbolic marker and have been proven as a strong approach to enhance group identification, group coordination, and in-group favoritism [49]. Third, each group participated in a 4-min online chat session to introduce themselves and identify

common features among group members. Brief conversations have been recognized as an effective way to establish and maintain social bond [88] and to strengthen interconnections among group members, particularly when the topic involved similarities or interdependence among them [89]. In contrast, participants in the no-bonding control session did not share color preference nor wear group uniform [15] and talked about their main courses without being explicitly asked to find shared features during the online communications. No verbal communication was allowed in both bonding and control sessions.

**Leader election and within-group interaction session.** Participants engaged in a 4-min within-group online chat session. Similar to previous studies [44,45], participants were asked to: (i) democratically elect a leader for their group; and (ii) discuss the strategies that their group would employ in potential intergroup contests. Participants had the autonomy to decide whether they preferred initiating discussions on strategies, electing a group leader first, or discussing both 2 topics in parallel. We made it clear to participants that leaders in the current experiment were elected rather than appointed and held symbolic roles without formal power or responsibility over other group members. During the online chatting, participants were labeled by different shapes and communicated via sending messages to remain anonymous (verbal communication was not allowed). For each three-person group, any group member could self-nominate or nominate their peers as the leader. Among all groups, 85% of groups ($N = 149$) nominated 1 candidate as the leader, who was then approved, while the remaining 15% of groups ($N = 27$) initially nominated multiple candidates but later identified one of them as the final leader later. Of all leaders, 37.5% ($N = 66$) self-nominated while 62.5% ($N = 110$) were nominated by their peers.

**Leader election validation.** All 176 three-person groups elected their leaders within the 4-min session and successfully recognized group leader in the post-survey. We conducted conversation analysis and compared the total number and length of the utterance given by leader and followers. We found that the group leader was more talkative than followers during group communications (total number of utterances: $t_{175} = 5.418$, $p = 1.970 \times 10^{-7}$, Cohen's $d = 0.408$, 95% CI: 1.315, 2.822; total length of utterances: $t_{175} = 8.316$, $p = 2.468 \times 10^{-14}$, Cohen's $d = 0.627$, 95% CI: 26.932, 43.693). Moreover, the group leader was more likely to initiate group interaction (leader: 40.3% versus three-person-average: 33.3%, $p = 0.039$). In addition, we asked participants to report their willingness to be the group leader (0 = not willing at all, 10 = extremely willing). We found that, both in bonding and control conditions, the group leader showed higher willingness and motivation to be the group leader (main effect of leader: $F_{1, 172} = 64.615$, $p = 1.40 \times 10^{-13}$, $\eta^2 = 0.273$). Moreover, leaders who underwent social in-group bonding exhibited a higher level of willingness to be the group leader (leader × bonding interaction: $F_{1, 172} = 0.067$, $p = 0.812$, $\eta^2 = 3.31 \times 10^{-4}$). Finally, the chat transcriptions of all 176 groups were evaluated by independent raters, who were blind to experimental hypotheses and conditions, along domains of (i) identifying the group leader of each group (identification accuracy = 97.66%); and (ii) reporting perceived interaction quality and leadership. These results suggested that leaders were not randomly elected or coerced into their positions and validated our experimental setup for establishing the roles of the group leader and followers in three-person groups.

**Economic games related to intergroup discrimination.** In the IDG, participants received an endowment of 100 monetary units (MUs) and decided how to split the 100 MUs between hypothetical in-group and out-group members. The different MUs to in-group and out-group members indicated the level of intergroup discrimination [15,58]. In the IPD-MDG, participants received an endowment of 100 MUs and decided how to split the 100 MUs among a self pool, a within-group pool, and a between-group pool [59,60]. Each MU to the self pool was kept as 1 MU to the participants themselves; each MU contributed to the within-group pool

added 0.5 MU to each in-group member, including the contributor, and each MU contributed to the between-group pool added 0.5 MU to each in-group member and in addition, subtracted 0.5 MU from each out-group member. Contributions to the within-group pool reflects "ingroup love" without hurting out-group members. Contributions to the between-group pool, in contrast, reflects spiteful "out-group hate."

## fNIRS data acquisition

Brain activity of 3 individuals of the same group was simultaneously recorded by 1 LABNIRS optical topography system (52-channel high-speed LABNIRS, Shimadzu Corporation, Japan). For each participant, 2 sets of homologous optode probes were used, with each measuring 7 channels (3 light emitters and 3 detectors, inter-optode distance of 30 mm, Fig 1B). The 2 probe sets separately covered the rDLPFC and the rTPJ, the position of which was based on the relevant standard positions of F4 and P6 in the 10–10 international system for electroencephalogram electrode placement (S8 Table, [90]). The anatomical localization of rDLPFC and rTPJ was confirmed in our previous study using high-resolution T1-weighted structural images [15].

The absorption of near-infrared light at 3 wavelengths (780 nm, 805 nm, and 830 nm) was measured at a sampling rate of 47.62 Hz and later down-sampled to 9.52 Hz by averaging 5 consecutive data points for all the analyses to decrease temporal autocorrelation [91]. The absorption changes were transformed into the relative concentration changes of oxygenated hemoglobin (oxy-Hb), deoxygenated hemoglobin (deoxy-Hb), and total hemoglobin based on the modified Beer–Lambert law [92]. It has been shown that the concentration changes of oxy-Hb are the most sensitive indicator of the regional cerebral blood flow in fNIRS measures and increases in oxy-Hb reflects the consequence of neural activity and corresponding to the blood oxygenation level dependent (BOLD) signal measured by fMRI [93]. Moreover, the oxy-Hb concentration has a better signal-to-noise ratio than deoxy-Hb [34]. Thus, similar to most previous fNRIS studies [15,34–36,40,44,45], the current study focused on the concentration changes of oxy-Hb.

## Behavioral data analysis

**Conversation analysis.** We analyzed the transcripts of turn-taking conversations for each three-person group. For the within-group communication of each group, we calculated: (i) the total number of utterances of group leader and followers, respectively; (ii) the number of turn-transition; and (iii) the turn response time (i.e., the time gap between turns) separately for inter-status and intra-status transitions. The inter-status turn-transition was defined as a discrete pair of a leader utterance followed by a follower utterance, or vice versa, whereas the intra-status turn-transition was defined as a follower utterance followed by another follower. The turn response time was then log transformed to avoiding skewness. To control for the total length of utterances of all 3 group members, we include the total word number as covariate in the further analysis. These conversation-related indices were submitted to two-way mixed-model ANCOVAs with a within-subjects factor hierarchy (inter-status versus intra-status or leader versus follower), a between-subjects factor bonding (bonding versus control), and total word number as covariate. It should be noted that, the mean turn response time in the current study deviated largely away from the verbal conversation turn response time of about 200 ms, possibly due to that group members communicated via online messaging rather than verbal communication. In addition, to mitigate the potential impact of displaying "typing" in the group chat window on response latency measurement, we have opted not to show status information (i.e., whether someone is typing or not) in the chat window. Only after

participants completed and submitted their messages into the group chat window, other members were then able to read others' messages.

**Participants' subjective ratings.** Participant's self-report of inter-status cohesion (i.e., cohesion between leader and follower) and intra-status cohesion (i.e., cohesion between followers) were submitted to a two-way mixed-model ANOVA with a within-subjects factor hierarchy (inter-status versus intra-status) and a between-subjects factor bonding (bonding versus control). Follower's ratings of perceived social influence and attraction of the group leader was compared between bonding and control conditions with two-tailed independent-sample $t$ tests. Moreover, we performed partial correlation analyses to test the relationship between the perceived social influence (attraction) and the turn transition (response time) separately for inter- and intra-status, controlling the utterance length of leaders and followers, respectively.

We next conducted mediation analysis to examine whether the effect of social bonding (the independent variable, $X$) on the leader's social influence or attraction (the dependent variable, $Y$) was mediated by social cohesion (the mediator, $M$). The mediation analyses were performed using PROCESS (model 4) in SPSS. For each mediation analysis, we computed both the indirect effect ($c$) and direct effect ($c'$), where $c$ measures the extent to which $X$ influences $Y$ through $M$, while $c'$ represents the effect of $X$ on $Y$ that cannot be explained by $c$. If $c$ is nonzero and $c'$ is zero, it indicates full mediation; otherwise, it suggests partial mediation. The significance of mediation was assessed using both Sobel test and Preacher–Hayes bootstrapping. The Sobel test was employed to determine if introducing $M$ significantly reduced the relationship between $X$ and $Y$. A two-tailed z-test was used to establish statistical significance level. On the other hand, bootstrapping method involved a nonparametric examination of whether zero fell within the bootstrapped 95% confidence intervals (5,000 times). If zero did not fall within the confidence intervals, we could conclude that there existed a significant indirect effect. It showed be noted that causal direction in the mediation analysis was not directly examined. Further research employing experimental and longitudinal approaches is warranted to validate our current findings from the mediation analysis.

**Intergroup discrimination.** Participants completed the IDG and IPD-MD before the experiment (as baseline) and at the end of the experiment. We were interested in the intergroup discrimination, i.e., differences in monetary allocation to in-group members and out-group members. Moreover, in IPD-MD, we were also interested in the amount of MU allocated to the within-group pool (i.e., "ingroup love") and the between-group pool (i.e., "outgroup hate"), respectively. We first compared the baseline intergroup discrimination measured before the experiment between bonding and control sessions and between leaders and followers. We found comparable baseline intergroup discrimination (main effect of bonding: $F_{1, 174} = 0.541$, $p = 0.463$, $\eta^2 = 0.003$; main effect of hierarchy: $F_{1, 174} = 0.017$, $p = 0.897$, $\eta^2 = 9.712 \times 10^{-5}$; bonding × hierarchy interaction: $F_{1, 174} = 0.007$, $p = 0.935$, $\eta^2 = 3.855 \times 10^{-5}$), ingroup love (main effect of bonding: $F_{1, 174} = 0.149$, $p = 0.700$, $\eta^2 = 0.001$; main effect of hierarchy: $F_{1, 174} = 0.257$, $p = 0.613$, $\eta^2 = 0.001$; bonding × hierarchy interaction: $F_{1, 174} = 1.484$, $p = 0.225$, $\eta^2 = 0.008$), and outgroup hate (main effect of bonding: $F_{1, 174} = 1.140$, $p = 0.287$, $\eta^2 = 0.007$; main effect of hierarchy: $F_{1, 174} = 0.447$, $p = 0.505$, $\eta^2 = 0.003$; bonding × hierarchy interaction: $F_{1, 174} = 0.320$, $p = 0.572$, $\eta^2 = 0.002$) among the 4 conditions. The intergroup discrimination in the IDG, the ingroup love and outgroup hate in the IPD-MDG, measured at the end of the experiment, were submitted to bonding × hierarchy (leader versus follower) ANOVAs to examine bonding effects on intergroup discrimination in leader and followers.

**Independent ratings on within-group interaction.** To objectively quantify the within-group interactions perceived by third-party observers, independent raters were asked to identify the leader of each group and leader emergence time and to evaluate the chatting messages of all the 176 groups along the following dimensions on a 9-point Likert scale: (i) interaction

frequency (1 = least frequent, 9 = most frequent); (ii) interaction intensity (1 = least intense, 9 = most intense); (iii) group cohesion (1 = least cohesive, 9 = most cohesive); and (iv) prominence of the group leader (1 = least prominent leader, 9 = most prominent leader). The independent raters correctly identified the group leader (accuracy: 97.66%). The ratings were reliable among the 14 raters (interaction frequency: ICC consistency: 0.940, agreement: 0.885; interaction intensity: ICC consistency: 0.814, agreement: 0.710; group cohesion: ICC consistency: 0.855, agreement: 0.757; leader prominence: ICC consistency: 0.813, agreement: 0.746). For each rating dimension, rating scores were first z-score transformed for each rater. The normalized ratings were then averaged across raters and compared between bonding and control conditions using two-tailed independent-sample $t$ tests. The rating data of one group was missing. We showed that social bonding indeed facilitated the outsider-perceived group interaction frequency ($t_{173} = 4.426$, $p = 1.696 \times 10^{-5}$, Cohen's $d = 0.669$, 95% $CI$: 0.264, 0.688, S3A Fig) and intensity ($t_{173} = 3.943$, $p = 1.166 \times 10^{-4}$, Cohen's $d = 0.596$, 95% $CI$: 0.157, 0.472, S3B Fig). Moreover, independent raters identified the leader of groups under bonding (versus control) condition faster ($t_{173} = -2.547$, $p = 0.012$, Cohen's $d = -0.385$, 95% $CI$: -40.522, -5.140, S3C Fig) and also recognized the prominence of group leader in the bonding condition to a greater degree than that in the control condition ($t_{173} = 2.001$, $p = 0.047$, Cohen's $d = 0.303$, 95% $CI$: 0.002, 0.339, S3D Fig).

## FNIRS data analysis

**Data quality check.**   The following steps were performed to check and ensure the data quality of fNIRS signals. First, prior to data recording, the resistance for each optode probe was monitored and adjusted to meet the minimum criteria defined in the LABNIRS recording software, aiming to achieve a good signal-to-noise ratio [94]. Second, during data recording, the raw optical density signal in each channel was transformed into concentration changes in HbO and HbR in real time, which were visually inspected to assess signal quality [95]. Third, during preprocessing of the fNIRS signals, we applied an automatic sliding-window detection to further evaluate data quality. Specifically, for each HbO and HbR time series, extreme values exceeding mean ± 3 standard deviations (SD) within a 10-s time-window were identified as outliers [96]. Channels with severe motion artifacts or containing more than 5% extreme values were labeled as bad. In order to maintain the same number of channels across all groups, any group with one or more bad channels was excluded from further analyses. Based on these criteria, a total of 8 groups were excluded from the final dataset due to poor data quality.

**Inter-brain neural synchronization (INS).**   We performed INS analysis on the neural data collected during the resting-state session (4 min, served as a baseline) and within-group interaction session (4 min). Similar to previous studies [15,34–37,44,45], we employed the WTC analysis to assess the cross-correlations between 2 oxy-Hb time series of dyads of participants as a function of frequency and time [97]. Specifically, within each three-person group, the 3 same-length oxy-Hb time series for each channel (i.e., oxy-Hb$_{leader}$, oxy-Hb$_{follower1}$, and oxy-Hb$_{follower2}$) were simultaneously acquired by the same fNIRS system. We applied WTC analysis to each pair of 3 oxy-Hb time series and generated 3 time-frequency matrices of the coherence values for each group (i.e., Coherence$_{leader-follower1}$, Coherence$_{leader-follower2}$, and Coherence$_{follower1-follower2}$). The coherence values from the leader-follower dyads (averaged value of Coherence$_{leader-follower1}$ and Coherence$_{leader-follower2}$) indicated the inter-status INS, and the coherence value from the follower–follower dyads (i.e., Coherence$_{follower1-follower2}$) indicated the intra-status INS. In each time-frequency matrix (Fig 3A), each row corresponded to a specific frequency point, each column corresponded to a specific time point and the color

bar corresponded to the coherence value. To ensure consistent data size for inter- and intra-status INS, we conducted a set of control analyses by calculating coherence values between the group leader and a randomly selected follower to index inter-status INS. With this approach, the data size and calculation of inter- and intra-status INS were matched. The results obtained from these control analyses replicated our main findings (S10 Fig).

fNIRS signals not only reflected task-evoked brain activity but also systemic physiological interference arising from cardiac pulsation (approximately 1 Hz), breathing rate (approximately 0.3 Hz), and other homeostatic processes [95]. Similar to previous studies [4,98], we employed the baseline subtraction approach to mitigate the impact of physiological noise. Specifically, we recorded a resting state session with an identical duration as the task session. As the resting-state predominantly reflects spontaneous hemodynamic oscillations [98], it served as the baseline for comparison. We performed WTC analysis on the neural data collected during both the resting-state session (4 min, serving as a baseline) and within-group interaction session (4 min). To reveal the effects of bonding and hierarchy on INS specific to group interaction, we focused on the increased INS during group interaction relative to the baseline (i.e., the resting-state). First, we compared coherence values (averaged across all channels and for each channel) between the within-group interaction session and resting-state session by performing paired-sample $t$ tests for each frequency (frequency range 0.01 to 1 Hz) [99] to identify the FOI (Fig 3A). This analysis identified increased coherence values in 2 frequency bands: between 0.136 Hz and 0.192 Hz (corresponding to the period between 5.20 and 7.35 s) and between 0.407 and 0.432 Hz (corresponding to the period between 2.31 and 2.46 s). These 2 frequency bands were chosen as the frequency of interest for the subsequent analyses (FOI, Bonferroni family-wise error (FWE) corrected for multiple comparisons). It is worth noting that no significant results were found in the frequency band of 0.407 to 0.432 Hz (full statistics were reported in S9 Table); therefore, only results based on 0.136 to 0.192 Hz were reported in the main text. This chosen period also effectively captures the temporal structure of the within-group interaction task since one-round within-group messaging typically took an average time of 5 to 7 s. In addition, this frequency band also excluded high- and low-frequency physiological noises, such as those related to respiration (about 0.2 to 0.3 Hz), cardiac pulsation (0.7 to 4 Hz), and high-frequency head movements (>1 Hz) [98].

We then calculated the session- and FOI-averaged coherence values and converted into Fisher z-scores. The increased INS (coherence differences between within-group interaction and resting-state) was submitted to bonding × hierarchy (inter-status versus intra-status) ANOVAs. Note that, the WTC algorithm normalized the amplitude of the signal within each time-window defined by the wavelet to make the data less vulnerable to transient spikes or motor artifacts [34–36]. Moreover, we conducted 2 sets of complementary analyses to further control the potential impact of physiological noises. First, we applied a wavelet-based denoising method to identify global physiological components per channel and extracted them out of the hemodynamic signals [15,61]. After the denoising process, the same WTC calculation and statistical analyses were applied, and the observed results were reserved (main effect of hierarchy at channel 3, $F_{1, 174} = 6.296$, $p = 0.013$, $\eta^2 = 0.035$, hierarchy × bonding interaction effect at channel 9, $F_{1, 174} = 5.311$, $p = 0.022$, $\eta^2 = 0.030$, S3A Table). Second, we controlled the globally co-varying signal using ANCOVA analyses [62], with the global mean INS (averaged coherence values across all channels) as the covariate. Significant results were fully replicated (main effect of hierarchy at channel 3, $F_{1, 174} = 9.220$, $p = 0.003$, $\eta^2 = 0.051$, hierarchy × bonding interaction effect at channel 9, $F_{1, 174} = 8.373$, $p = 0.004$, $\eta^2 = 0.046$, S3B Table).

**Time-lagged analysis.** To investigate the directionality of the inter- and intra-status INS, we conducted time-lag analysis [63–65] for the channels that showed increased INS during the within-group interaction stage, i.e., channel 3 in the rTPJ and channel 9 in the rDLPFC. For

each leader–follower (or follower–follower) dyad, the time courses of neural activity of the leader (1 follower) were shifted relative to that of the follower (the other follower) from −10 to 10 s (in 1-s increment). We then recalculated the inter- and intra- status INS on each time lag for both resting-state and within-group interaction. The time-lagged inter- and intra-status neural alignment increases (i.e., lagged INS during within-group interaction minus that during resting) on each time lag were compared with 0 using one-sample $t$ tests and compared between bonding and control conditions using two-tailed independent-sample $t$ tests. Significant effects were thresholded at $p < 0.05$, FDR corrected for multiple comparisons of the 21 time lags. Next, we performed correlation analysis between the neural alignment on each time lag and inter-group discrimination, for inter-status and intra-status dyads, respectively, the correlation coefficients were FDR corrected for multiple comparisons of the 21 time lags. We also performed Pearson's correlation coefficient analysis separately for bonding and control conditions.

**Permutation test.** First, we aimed to validate the INS increases (i.e., group interaction versus resting-state) in real groups. To this end, we examined which conditions showed increased INS (i.e., significant INS increases during group interaction compared to resting-state). Specifically, we compared INS differences (group interaction minus resting-state) against zero for the channels that exhibited significant effects of interest (i.e., channel 3 in the rTPJ and channel 9 in the rDLPFC). We found increased INS for the inter-status dyads at channel 3 in the rTPJ under control condition ($t_{1,86} = 3.943$, $p = 1.64 \times 10^{-4}$) and bonding condition ($t_{1,88} = 3.394$, $p = 0.001$; survived multiple correction) and at channel 9 in the rDLPFC (bonding: $t_{1,88} = 3.378$, $p = 0.001$; survived multiple correction). We then performed permutation test to examine whether these conditions showed increased INS in real group than pseudo groups. Specifically, within each condition, we generated three-person pseudo-groups by randomly grouping 3 participants from different original real groups. For each pseudo-group, the INS of each dyad was recalculated. This procedure was repeated for 1,000 times to generate a pseudo-group INS distribution. The increased INS in the aforementioned conditions of real groups were compared with each condition-specific permutation distributions. The significance level, $p$-value was indicated as: $p = j/1,000$, where $j$ is the number of samples out of the 1,000 permutation samples, of which the examined value was larger than the observed value of real groups. The results indicated that the increases in inter-status INS in the rTPJ (control: $p = 0.022$; bonding: $p = 0.047$) and rDLPFC (bonding: $p = 0.032$) of real groups all exceeded the upper 95% CI of the permutation distribution. These findings further confirmed an increased INS in these specific conditions within real (rather than random) groups.

Next, to validate the observed bonding and/or hierarchy effects on INS, we performed another 2 sets of permutation tests: (i) the within-condition permutation test; and (ii) cross-condition permutation test. First, within the bonding and control conditions, leader and followers of the real groups were randomly reassigned into new groups to form three-person pseudo groups. We then recalculated the inter- and intra-status INS for the 176 pseudo groups. This shuffling and recalculation procedures were repeated 1,000 times to generate permutation distributions for the observed hierarchy effect in rTPJ and bonding × hierarchy interaction effect in the rDLPFC. We then compared the observed effects of real interacting groups against 1,000 permutation samples and examined whether real effect exceeded the upper limits 95% or 99% CI of the permutation distribution. Second, similar procedures and statistical analyses were conducted for the cross-condition permutation test except that we generated cross-condition, three-person pseudo groups by randomly grouping 1 leader and 2 followers across bonding and control conditions as a pseudo-group.

**rDLPFC-rTPJ functional connectivity.** We applied cross-correlation analysis using the Functional Connectivity Toolbox [100] implemented in MATLAB to assess the functional connectivity of each rDLPFC-rTPJ channel pair (49 channel pairs, 7 channels in TPJ, and 7

channels in DLPFC) for each participant, which was then Fisher z transformed [101]. We also averaged the 49 channel pairs to index the grand mean of functional connectivity. The channel-pairwise connectivity and grand mean connectivity were separately submitted to bonding × hierarchy (leader versus follower) ANOVAs (FDR correction for multiple comparisons of 49 channel pairs was applied to channel-pairwise analysis). Next, the channel showing significant bonding effect on leader-to-follower neural alignment (i.e., CH9 in the rDLPFC) was used as the seed channel and the averaged functional connectivity across the 7 CH9-rTPJ channel pairs was used to index the channel-based functional connectivity. We then conducted correlation analysis between the channel-based functional connectivity and the leader-to-follower neural alignment in rDLPFC.

**Additional analyses and results.** To test whether the effects of social in-group bonding on hierarchical interaction were modulated by different interaction stages, we conducted additional analyses. First, we identified the specific time point at which the group leader explicitly emerged in each group. This analysis revealed that the establishment of a group leader occurred approximately halfway through the within-group interaction (with a mean value of 145.60 s, SE = 5.76). Second, based on this identified time point for leader emergence, we divided the within-group interaction session into 2 stages: pre- and post-leader emergence stages. Third, we conducted 3-way ANOVAs to examine whether these interaction stages influenced our main findings regarding hierarchy and/or bonding effects on both behavioral and neural indices. The results showed no significant impact of the interaction-stage on our main findings (S10A Table). Furthermore, despite observing an average occurrence of leader emergence around the middle of the within-group interaction period, we further balanced the duration between pre- and post-leader emergence stages by dividing it equally into 2 parts for subsequent analyses. This mid-split approach replicated our observation of no significant impact of different interaction stages (S10B Table). These results suggested that the effects of social bonding on hierarchical interaction remained consistent across different stages of within-group interaction. Alternatively, it is possible that in the current experimental setting, the group leader implicitly emerged prior to the explicit emergence time point (e.g., during the bonding section). Supporting this possibility, we observed that the group leader was more likely to initiate group interaction at the beginning of within-group interaction and was already more talkative in the pre-emergence stage (total number of utterances: $t_{175} = 4.877$, $p = 2.404 \times 10^{-6}$; total length of utterances: $t_{175} = 7.451$, $p = 4.076 \times 10^{-12}$). Furthermore, a majority of groups (85%, $N = 149$) early on nominated the individual who later emerged as their leader. We encourage future studies to directly investigate these possibilities. In addition, it is important to note that this examination of time effect was merely an initial exploration conducted at a coarse time scale. Thus, the neural dynamics at second or millisecond resolution needs to be directly and systematically examined in future studies.

To present the measured brain activity from different perspectives, we repeated our analysis on the deoxygenated hemoglobin signals (HbR). Specifically, we calculated the INS and intra-brain rDLPFC-rTPJ functional connectivity on HbR signals. We then performed the bonding × hierarchy ANOVAs on HbR-INS and HbR-FC to examine whether the main findings obtained with HbO signals would be similarly observed with HbR signals. Similar to the pattern of the HbO signals, we observed a significant, although weaker (did not survive multiple corrections), hierarchy effect in rTPJ (channel 3, $F_{1, 174} = 4.736$, *uncorrected p* = 0.031, $\eta^2 = 0.026$, S11 Table and S11A Fig), with stronger inter-status INS than intra-status dyads. However, no bonding × hierarchy interaction effect was observed on HbR signals (channel 9, $F_{1, 174} = 0.420$, $p = 0.518$, $\eta^2 = 0.002$). The difference observed on INS based on HbO and HbR signals may be caused by different sensitivities of these 2 types of signals in reflecting task-induced changes in neural signals. Regarding the intra-brain FC index, the leader effect on the

rDLPFC-rTPJ functional connectivity (stronger in leader than followers) based on HbO signals was similarly observed in the HbR-FC analysis (48 rDLPFC-rTPJ channel pairs survived FDR correction for 49 channel pairs, S12 Table; S11B Fig for the grand mean rDLPFC-rTPJ connectivity, $F_{1, 174} = 27.345$, $p = 4.842 \times 10^{-7}$, $\eta^2 = 0.136$).

## Statistical analysis

Similar to previous studies [4,15,102], data were aggregated at the three-person group level and hierarchy within each group (i.e., leader versus follower, or inter-status versus intra-status) were treated as a within-subjects factor. For both the behavioral and neural data, we averaged the 2 followers or the 2 leader–follower dyads to index the follower or inter-status level. The experimental condition (bonding versus control) was randomly introduced and blinded to the participants during data collection. For each dependent variable, the three-person groups whose value was larger or smaller than 5 SDs from the mean value were excluded. This data cleaning procedure led to exclusion of data in the following variables: intra-status turn transition ($n = 1$), intra-status turn response time ($n = 1$), and inter-status turn response time ($n = 2$). Two-way mixed-model ANOVAs were conducted on final behavioral and neural datasets with bonding (bonding versus no-bonding control) as a between-subjects factor and hierarchy (inter-status versus intra-status, or leader versus follower) as a within-subjects factor. Furthermore, the LMM is another optimal method for analyzing such structural data. Therefore, we performed a series of LMM analyses on behavioral and neural indices, considering hierarchy and bonding as fixed effects while treating each group as a random effect. This set of analyses yielded similar results to ANOVA.

ANOVA with significant interaction were followed by planned two-tailed $t$ tests to examine: (i) bonding effects (two-tailed independent-sample $t$ test) separately on inter-status and intra-status dyads or leaders and followers; and (ii) hierarchy effects (two-tailed paired-sample $t$ test) separately in bonding and control condition. Statistical significance was thresholded at $p < 0.05$. Data distributions were assumed to be normal, but this was not formally tested. For ratings from independent sample, data were first normalized across items for each rater and then averaged across all raters. Correlation analyses were conducted using Pearson's correlation coefficient analysis. It should be noted that the reported behavior-neural correlations, although statistically significant, should be interpreted and applied with caution due to their small to medium effect sizes [103]. All statistical analyses were performed with SPSS (IBM SPSS Statistics 25) and custom scripts in MATLAB (R2017b & R2020b, The MathWorks, United States of America). The wavelet coherence analysis was performed by Wavelet Coherence Package [104] implemented in MATLAB (which is available in https://noc.ac.uk/business/marine-data-products/cross-wavelet-wavelet-coherence-toolbox-matlab).

## Supporting information

**S1 Data. Numerical data underlying graphs.** Names of individual sheets correspond to figure panels for which the numerical data is used: **Figs 1D–1H,** 2A–2E, 3B–3H, 4B–4E and 5A–5C. (XLSX)

**S1 Fig. Experimental procedure.** Before coming to the laboratory, participants completed an online survey that included demographic and psychological information as well as color preference (white versus black, for bonding manipulation). One to 4 days later, participants were invited to the laboratory in groups of 3 same-gender strangers and randomly assigned into either the bonding or control condition. They were instructed to sit face-to-face in a triangle and completed 3 sessions during fNIRS-based hyper-scanning: (i) a 4-min resting-state

session; (ii) a 4-min in-group social bonding (or no-bonding control) manipulation session; and (iii) a 4-min online within-group interaction session. At the end of the experiment, participants were asked to complete a series of intergroup-related economic games (including intergroup dictator game and intergroup prisoner's dilemma-maximizing differences game), as well as report subjective evaluations on group cohesion, leader influence and attraction, positive attitudes, willingness to become the leader, etc. (details in Methods).
(TIF)

**S2 Fig. In-group social bonding facilitates the within-group communication.** Social bonding particularly increased the utterance numbers given by leaders (control: 8.590 ± 4.088, bonding: 12.750 ± 5.911) than followers (control: 7.322 ± 3.655, bonding: 9.899 ± 4.311). Data are plotted as box plots for each condition, with horizontal lines indicating median values, boxes indicating 25% and 75% quartiles and whiskers indicating the 2.5%–97.5% percentile range. Cross symbols in each box represent the mean values. Data points outside the range are shown separately as circles. $^*p < 0.05$, $^{***}p < 0.001$.
(TIF)

**S3 Fig. Bonding effect on within-group communication was perceived by third-party observers.** Social bonding increased the perceived group interaction frequency (control: −0.237 ± 0.787, bonding: 0.240 ± 0.629, **A**) and intensity (control: −0.158 ± 0.542, bonding: 0.156 ± 0.512, **B**). Third-party observers identified the group leader faster (control: 189.988 ± 49.554, bonding: 167.157 ± 67.349 **C**), and perceived the leader as more prominent (control: −0.0095 ± 0.582, bonding: 0.076 ± 0.547, **D**) in the bonding condition. Data are plotted as box plots for each condition, with horizontal lines indicating median values, boxes indicating 25% and 75% quartiles and whiskers indicating the 2.5%–97.5% percentile range. Cross symbols in each box represent the mean values. Data points outside the range are shown separately as circles. $^*p < 0.05$, $^{***}p < 0.001$.
(TIF)

**S4 Fig. The effect of in-group social bonding on leader behavior and the perception of leader.** (**A**) Under social bonding, followers perceived greater social attraction of the leader (control: 6.333 ± 2.036, bonding: 7.135 ± 1.548). (**B**) Leader's social attraction was positively associated with inter-status cohesion (Pearson's correlation analysis). Each solid line represents the least squares fit, with shading showing the 95% *CI*. (**C/D**) Leader's social influence (**C**) and attraction (**D**) were positively associated with intra-status cohesion. (**E**) Bonding increased perceived social attraction of the leader through enhancing inter-status cohesion. $^*p < 0.05$, $^{**}p < 0.01$, $^{***}p < 0.001$.
(TIF)

**S5 Fig. Validation of INS results by nonparametric permutation tests.** (**A/B**) We generated within-condition pseudo-groups by randomly grouping a real leader and 2 real followers from different original groups in the same bonding or control condition to 1 pseudo-group (**A**), or generate across-condition pseudo-groups by randomly grouping 1 leader and 2 followers across bonding and control conditions as one pseudo-group (**B**). The inter- and intra-status INS for each pseudo group were recalculated. These procedures were repeated for 1,000 times to generate permutation distributions. (**C/D**) We compared the hierarchy main effect in the rTPJ and the interaction effect in the rDLPFC of real group against cross-condition permutation distributions ($n = 1,000$). The observed effects of Hierarchy in the rTPJ (**C**) and of Hierarchy × Bonding interaction in the rDLPFC (**D**) exceeded the upper limits of 99% *CI* of the permutation distributions.
(TIF)

**S6 Fig. Inter-status neural alignment in the rTPJ was significant in both directions.** (**A**) Inter-status neural alignment in rTPJ is significant from −10 to +10 time lags (peaked at 0 s), survived FDR multiple correction. The significant time lags (survived multiple correction) are highlighted with the horizontal line on the x-axis. (**B**) The inter-status neural alignment in rTPJ is significant at all time lags in bonding and control conditions separately. Shaded areas represent standard error (SE).
(TIF)

**S7 Fig. Leader-to-follower neural alignment in rDLPFC was positively associated with intergroup discrimination.** (**A–F**) Correlation analyses between leader-to-follower neural alignment at each time lag (+1 to +6) with intergroup discrimination. Correlations were performed by Pearson's correlation coefficient analysis. Each solid line represents the least squares fit, with shading showing the 95% *CI*. † $p < 0.06$, *$p < 0.05$, **$p < 0.01$.
(TIF)

**S8 Fig. No significant bonding effect or correlation on intra-status neural alignment.** (**A**) In-group social bonding showed no significant effect on intra-status neural alignment in rDLFPC at any time lags. (**B**) Intra-status neural alignment in rDLFPC did not correlate with intergroup discrimination at any time lags.
(TIF)

**S9 Fig. Stronger rDLPFC-rTPJ functional connectivity in leaders accounted for leader-to-follower neural alignment.** (**A–F**) Correlation analyses between leaders' rDLPFC-rTPJ connectivity with leader-to-follower neural alignment at each time lag (+1 to +6). Correlations were performed by Pearson's correlation coefficient analysis. Each solid line represents the least squares fit, with shading showing the 95% *CI*. † $p < 0.07$, *$p < 0.05$.
(TIF)

**S10 Fig. Neural synchronization between the group leader and a randomly selected follower.** The significant hierarchy main effect in the rTPJ (**A**) and bonding × hierarchy interaction effect in the rDLPFC (**B**) were fully replicated when considering inter-status INS for the leader and a randomly selected follower (channel 3, $F_{1, 174} = 8.330$, $p = 0.004$, $\eta^2 = 0.046$, **A**; channel 9, $F_{1, 174} = 11.133$, $p = 0.001$, $\eta^2 = 0.060$, **B**). Data are plotted as box plots for each condition, with horizontal lines indicating median values, boxes indicating 25% and 75% quartiles and whiskers indicating the 2.5%–97.5% percentile range. Cross symbols in each box represent the mean values. Data points outside the range are shown separately as circles. **$p < 0.01$.
(TIF)

**S11 Fig. Inter-brain neural synchronization and intra-brain functional connectivity in HbR signals.** (**A**) HbR-INS in rTPJ showed a significant, but weaker (did not survive multiple corrections), hierarchy main effect (channel 3, $F_{1, 174} = 4.736$, *uncorrected* $p = 0.031$, $\eta^2 = 0.026$), with stronger inter-status INS than intra-status one. (**B**) The grand mean of rDLPFC-rTPJ connectivity in HbR exhibited a significant hierarchy main effect ($F_{1, 174} = 27.345$, $p = 4.842 \times 10^{-7}$, $\eta^2 = 0.136$), with stronger rDLPFC-rTPJ functional connectivity in leaders (vs. followers). Data are plotted as box plots for each condition, with horizontal lines indicating median values, boxes indicating 25% and 75% quartiles and whiskers indicating the 2.5%–97.5% percentile range. Cross symbols in each box represent the mean values. Data points outside the range are shown separately as circles. ***$p < 0.001$.
(TIF)

**S1 Table. Demographic and psychological information of participants.**
(DOCX)

**S2 Table. Full statistical reports of the results of Hierarchy × Bonding mixed-model ANOVAs on inter-brain neural synchronization.**
(DOCX)

**S3 Table. Statistical reports of inter-brain neural synchronization in 2 complementary analyses.** (A) ANOVA analysis after the wavelet-based denoising. (B) ANCOVA analysis controlling global mean INS.
(DOCX)

**S4 Table. Full statistical reports of the results of bonding effect (Bonding vs. Control) on inter-status neural alignment (CH9) for each time lag.**
(DOCX)

**S5 Table. Full statistical reports of the results of inter-status INS increase (CH9, one-sample *t* tests) of each time lag under bonding and control conditions, respectively.**
(DOCX)

**S6 Table. Full statistical reports of the results of the Pearson's correlations between inter-status neural alignment (CH9) and intergroup discrimination on each time lag.**
(DOCX)

**S7 Table. Full statistical reports of hierarchy main effect on rDLPFC-rTPJ functional connectivity for each channel pair.**
(DOCX)

**S8 Table. The anatomical position for each recording channel.**
(DOCX)

**S9 Table. Full statistical reports of the results of Hierarchy × Bonding mixed-model ANOVAs on inter-brain neural synchronization in frequency band 0.407–0.432 Hz.**
(DOCX)

**S10 Table. Statistical reports of interaction-stage/time-bin modulation effects on main findings. (A)** Analysis on pre- and post-leader emergence stages (based on leader emergence time). **(B)** Analysis on early and late time-bin (equally split by mid-point).
(DOCX)

**S11 Table. Full statistical reports of the results of Hierarchy × Bonding mixed-model ANOVAs on HbR-INS.**
(DOCX)

**S12 Table. Full statistical reports of hierarchy main effect on rDLPFC-rTPJ HbR-FC for each channel pair.**
(DOCX)

## Acknowledgments

We thank H. Zhang, C. Yang, and X. Zou for their assistance in data collection.

## Author Contributions

**Conceptualization:** Yina Ma.

**Data curation:** Jun Ni, Jiaxin Yang.

**Formal analysis:** Jun Ni.

**Funding acquisition:** Yina Ma.

**Investigation:** Jiaxin Yang, Yina Ma.

**Methodology:** Jun Ni, Yina Ma.

**Supervision:** Yina Ma.

**Visualization:** Jun Ni, Yina Ma.

**Writing – original draft:** Jun Ni, Yina Ma.

**Writing – review & editing:** Jun Ni, Yina Ma.

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
