## [Editor Report · Decision Letter 0]

15 Aug 2023

Dear Dr Ma, 

Thank you for submitting your manuscript entitled "Social bonding shapes hierarchical interaction and neural alignment 

in human groups" for consideration as a Research Article by PLOS Biology.

Your manuscript has now been evaluated by the PLOS Biology editorial staff as well as by an academic editor with relevant expertise and I am writing to let you know that we would like to send your submission out for external peer review.

Once your full submission is complete, your paper will undergo a series of checks in preparation for peer review. After your manuscript has passed the checks it will be sent out for review. To provide the metadata for your submission, please Login to Editorial Manager (https://www.editorialmanager.com/pbiology) within two working days, i.e. by Aug 17 2023 11:59PM.

Kind regards,

Christian

Christian Schnell, PhD

Senior Editor

PLOS Biology

cschnell@plos.org

---

## [Decision Letter · Decision Letter 1]

20 Oct 2023

Dear Dr Ma,

Thank you for your patience while your manuscript "Social bonding shapes hierarchical interaction and neural alignment in human groups" was peer-reviewed at PLOS Biology. It has now been evaluated by the PLOS Biology editors, an Academic Editor with relevant expertise, and by several independent reviewers. 

In light of the reviews, which you will find at the end of this email, we would like to invite you to revise the work to thoroughly address the reviewers' reports.

As you will see below, the reviewers think that the study is well executed and provides important insights but ask for a couple of clarifications and additional analyses to support the conclusions.

Given the extent of revision needed, we cannot make a decision about publication until we have seen the revised manuscript and your response to the reviewers' comments. Your revised manuscript is likely to be sent for further evaluation by all or a subset of the reviewers.

**IMPORTANT - SUBMITTING YOUR REVISION**

*Re-submission Checklist*

*Published Peer Review*

*PLOS Data Policy*

*Blot and Gel Data Policy*

Sincerely,

Christian

Christian Schnell, PhD

Senior Editor

PLOS Biology

cschnell@plos.org

REVIEWS:

Reviewer #1: The manuscript reports an impressive study on the effects of a social bonding intervention on hierarchical interactions in small groups of three people. The topic is very interesting and timely. A great strength of the study is the large data set that was collected and the analysis of both intra- and inter-status interaction patterns on the behavioral and neural levels. 

I have some questions and suggestions to further improve the manuscript:

The manuscript would benefit from clear theory-driven hypotheses. Did the authors expect to find differential effects of the bonding manipulation on intra- vs. inter-status interactions? If yes, these would be good to delineate. If the study was more explorative in nature, this is fine as well, but it should be made explicit. 

Relatedly: Was there a pre-registration? If yes, this should be made public and linked to the manuscript. If not, this should be mentioned.

Please briefly mention the type of social bonding manipulation that was used in the introduction. I kept wondering, but I did not want to switch back and forth between the sections. A sentence or two will suffice.

Why were fNIRS recordings only performed over participants' right hemispheres? A clearer rationale would be helpful.

A figure outlining the sequence of tasks and measures would be helpful for the reader.

The authors report only HbO data and try to justify their choice in the methods section. Yet, it is good practice to also report HbR data in fNIRS research. They should add corresponding analyses, at the very least as additional analyses in the supplements. If the same patterns are found as with HbO, that's great. If no significant results can be found that is also informative for other researchers and it could support the authors' notion that HbO is the more sensitive measure. 

Permutation analysis: Maybe I missed this, but did the authors find significantly increased WTC in real groups vs. random groups? That should be stated.

In the discussion, I would be interested to learn more about the authors' thoughts on the underlying socio-cognitive mechanisms of how exactly the bonding manipulation influenced inter-status interaction patterns, in particular.

Conversation analysis: Did participants see when another person was typing? Was this indicated in the chat window? I would imagine that this has quite a strong effect on the response latencies as - different from vocal conversation - it is in principle possible to formulate a response simultaneously in written conversation.

INS: Analysis: There was twice as much data included in the WTC for inter-status than intra-status couples. Was this taken into consideration in the statistical analysis?

Information on the fNIRS pre-processing seems to be missing. Please explain your pre-processing steps, including motion correction, data quality checks, exclusion of noisy channels, filtering etc. 

Were the independent raters ignorant of condition/ study hypotheses?

Statistical analyses: I was wondering why the authors chose to mainly work with ANOVAs. Would the nested data structure not be better served with LMM?

Minor:

I understand that the authors conducted the study in one cultural context, and it is a great starting point to have collected such an impressive dataset. Yet, I was wondering whether cultural context and culture-specific values might play a role for hierarchical group interactions, especially differential patterns in intra- and inter-status interactions. Perhaps this could be mentioned as an outstanding question for future research in the discussion?

Reviewer #2: The current manuscript explores the effects of social bonding on small hierarchical groups, using a combination of behavioral, neuroimaging, and computational methods. The authors recruited 176 participants to form 58 groups of three individuals each, and manipulated social bonding between group members using a color preference task. They then had participants engage in a series of tasks designed to measure group interaction quality, including a group decision-making task and a group competition task. 

The current study uniquely combines a set of behavioral manipulations, fNIRS neuroimaging, and applied computational methods with various validations of the findings. The most surprising here is the large sample size, resulting in robust effects. In addition, I must note the clear and well-organized presentation of results, which makes it easy for readers to follow the authors' arguments.

In my opinion, this is one of the best hyperscanning studies that I ever seen!

It should contribute to our understanding of the real-time neurocognitive mechanisms underlying social bonding and may result in further implications for improving team-building and group dynamics in various social contexts.

A feedback for improvements:

Lines 110-111: the authors discuss very briefly previous hyperscanining studies, and the transition to your study looks too sharp. Previous studies have tried to investigate symmetric and non-symmetric social communications. Also, some of the paradigms are close to the current study. For example, you can discuss teacher-teacher vs. teacher-student interactions (e.g. https://doi.org/10.1016/j.bandc.2021.105803), manager-employer interactions (e.g. https://doi.org/10.7358/neur-2019-026-bal2) or maybe you will find even better fit. This way, the transition to the current study should be smoother.

At the same location (lines 110-111), the authors focus the whole discussion just on the prefrontal cortex. However, IBS in other regions was reported (e.g. rTPJ) and the authors admitted it later. Thus, this part is confusing and not synced with the latter.

You are using only 4-minute length bonding tasks. It souls like to short for social bonding. Please clarify your choice.

Line 875: Please specify what kind of family-wise error rate is used.

The authors provide a very detailed description of the statistical methodology for all analyses except the mediation. We should always remember that statistical testing of the mediation is just based on the pattern of correlations in the data. However, the mediation model assumes a causal relationship. Did you try to create any quasi-casual effect using lagging between the mediator and the outcome?

Third-party observers' validation of the interactions is very impressive. I don't think that will make any impact on the results, but inter-rater agreement should be estimated with intra-class correlation (ICC).

Status-group interactions (2X2). Using 4 colors is pretty complex for reading the graphs. Maybe using just 2 colors (in addition to the x-axis) should be good enough.

While the authors did a good job of discussing the potential implications of their findings for team-building and group dynamics, it would be helpful to have a more detailed discussion of the potential limitations.

 Are the current findings biased by the cultural context? 

Most of the correlations between behavioral and neural variables are very low. They are statistically significant due to the large sample size but should be interpreted with caution. 

For future studies, it could be great to look at the time trajectories of the behavioral and neural data. Such an approach may provide more insights into the mechanistic characteristics of IBS and social bonding, exploring non-linear effects and running mediation analyses.

Minor: It looks like the color legend for 3B appears in 3C.

Reviewer #3: Overall, this is an interesting paper that uses rigorous and inventive methods to reveal an intriguing set of results. Hopefully, addressing the questions and concerns below can further strengthen this paper and resolve some potential issues.

1) In order to improve readers' ability to both understand and assess the findings and potential interpretations thereof, it would be helpful to include more information on key aspects of the the paradigm earlier on - such as how important concepts, such as "social bonding" and "leader" were operationalized.

For example, given that "social bonding" encompasses many possible experiences and activities, it would be helpful to provide more context regarding what "social bonding" entailed in the current study earlier in the paper (e.g., at the end of the introduction, when discussing the current work). Even though the Methods section is at the end of the paper at this journal, this aspect of the paradigm seems very important to clarify to readers to provide context when reading the results. Similarly, it would be helpful to provide more context on this in the paper's abstract.

On a related note, it would be helpful to include more information on how leaders were selected and what differentiated their role. What topics did participants discuss in the online chat when determining who would be a group leader? Did the participants have to discuss the given topic before selecting a leader or could they decide on a leader, then discuss the topic? Was it typically the case that no group member wanted to be a leader and someone was selected at random or asked to be a leader by group members, or was it typically the case that multiple group members wanted to be the leader? 

2) Clarifying some of the questions mentioned above about the group interaction/leadership selection phase seem to be very important for understanding the paradigm, for informing interpretations of the results, and critically, for evaluating the appropriateness of the paper's overall framing. The participants have 4 minutes to discuss a given topic and elect a leader of their group, and neural synchrony was assessed during this "group interaction" phase. It seems very important to distinguish between neural synchrony results that pertain to the process of leadership selection vs. conversations among group members after a leader is selected. Thus, it would be helpful to separately analyze these conversations. If some or most of the group interaction time that was analyzed pertain to time *before* a leader was selected and/or during the leader selection process itself, rather than interactions among followers and already-established leaders, this would very significantly change the interpretation of the results and the way that the paper should be framed overall.

3) The time-lagged analyses are especially interesting. However, it is not necessarily the case that the group leaders' brain activity preceding that of followers reflects something that the leader is doing (e.g., "anticipating and predicting the mental stats of followers"), rather than something that the followers are doing (e.g., effectively following the leader). However, this seems to be the only interpretation offered and this interpretation is offered several times. Couldn't it also be that following social bonding, followers are particularly attentive to group leaders, and thus, better at "following" them? The pattern of results shown here does not seem to necessarily entail predictive processing on the part of the leaders.

4) Relatedly, on p. 20, the authors state, "The functional connectivity between rDLPFC and rTPJ has been shown to play a key role in perspective taking, mental inference, and information integrating [74, 75]. We thus expected stronger rDLPFC-rTPJ functional connectivity in leaders than followers, which might account for the leader-to-follower neural alignment." Why would one not have expected followers to also need to engage in perspective taking, mental inference, and information integration? Was this prediction made a priori and if so, what was it based on?

Minor points:

1) It would be pertinent to discuss the extent to which the current findings would be expected to generalize across contexts and tasks. Relatedly, to what extent do the processes of selecting leaders and exerting leadership here resemble how leadership works outside of the lab? More generally, a discussion of ecological validity would be important.

2) Proofreading would be helpful throughout the manuscript to fix typos (e.g., "In additional") and enhance clarity. 

3) No demographic information about participants seems to be provided. It would be helpful to include this. 

4) On a related note, it would enrich the paper to discuss how the current patterns of results might be expected to differ or generalize across cultures and contexts..

5) It appears that the researchers gathered people in-person to do a totally online study. It was a bit confusing when initially reading the paper to discern what parts were in-person and what parts were online, so this would be helpful to clarify early on in the paper.

6) The rationale for and significance of conversation characteristics (e.g., turn-response time, turn-transition) might not be clear to all readers. Therefore, it could be helpful to discuss this a bit more.

7) It would be helpful to discuss the rationale for focusing on the right side of the brain.

---

## [Decision Letter · Decision Letter 2]

30 Jan 2024

Dear Dr Ma,

Thank you for your patience while we considered your revised manuscript "Social bonding shapes hierarchical interaction and neural alignment in human groups" for publication as a Research Article at PLOS Biology. This revised version of your manuscript has been evaluated by the PLOS Biology editors, the Academic Editor and the original reviewers.

Based on the reviews and on our Academic Editor's assessment of your revision, we are likely to accept this manuscript for publication, provided you satisfactorily address the following data and other policy-related requests.

* We would like to suggest a different title to improve readability: "Social bonding in groups of humans selectively increases information exchange and prefrontal neural synchronization"

* Please add the links to the funding agencies in the Financial Disclosure statement in the manuscript details

* In the ethics statement, please provide information about how how the participants provided consent to participate in the study: written or oral?

We expect to receive your revised manuscript within two weeks. 

*Published Peer Review History*

*Press*

Sincerely,

Christian

Christian Schnell, PhD

Senior Editor

cschnell@plos.org

PLOS Biology

Reviewer remarks:

Reviewer #1: The authors have done a very good job addressing my previous questions and recommendations. I don't have further comments.

Reviewer #2: I'm fully satisfied with the responses and looking forward to the publication of the manuscript in Plos Biology

---

## [Editor Report · Decision Letter 3]

12 Feb 2024

Dear Yina,

Thank you for the submission of your revised Research Article "Social bonding in groups of humans selectively increases inter-status information exchange and prefrontal neural synchronization" for publication in PLOS Biology. On behalf of my colleagues and the Academic Editor, David Poeppel, I am pleased to say that we can in principle accept your manuscript for publication, provided you address any remaining formatting and reporting issues. These will be detailed in an email you should receive within 2-3 business days from our colleagues in the journal operations team; no action is required from you until then. Please note that we will not be able to formally accept your manuscript and schedule it for publication until you have completed any requested changes.

PRESS

Sincerely, 

Christian

Christian Schnell, PhD, 

Senior Editor

PLOS Biology

cschnell@plos.org